# Budgeted Online Continual Learning by Adaptive Layer Freezing and Frequency-based Sampling

**Minhyuk Seo**[*]   **Hyunseo Koh**[*]   **Jonghyun Choi**[†]
Seoul National University
minhyukseo@yonsei.ac.kr, khs8157@gmail.com, jonghyunchoi@snu.ac.kr

## Abstract

The majority of online continual learning (CL) advocates single-epoch training and imposes restrictions on the size of replay memory. However, single-epoch training would incur a different amount of computations per CL algorithm, and the additional storage cost to store logit or model in addition to replay memory is largely ignored in calculating the storage budget. Arguing different computational and storage budgets hinder fair comparison among CL algorithms in practice, we propose to use floating point operations (FLOPs) and total memory size in Byte as a metric for computational and memory budgets, respectively, to compare and develop CL algorithms in the same 'total resource budget.' To improve a CL method in a limited total budget, we propose adaptive layer freezing that does not update the layers for less informative batches to reduce computational costs with a negligible loss of accuracy. In addition, we propose a memory retrieval method that allows the model to learn the same amount of knowledge as using random retrieval in fewer iterations. Empirical validations on the CIFAR-10/100, CLEAR-10/100, and ImageNet-1K datasets demonstrate that the proposed approach outperforms the state-of-the-art methods within the same total budget. Furthermore, we validate its effectiveness in the Multi-modal Concept incremental Learning setup with multimodal large language models, such as LLaVA-1.5-7B. Code is available at https://github.com/snumprlab/budgeted-cl.

## 1 Introduction

In a realistic scenario for continual learning (CL), data arrive in a streaming manner, prompting interest in online CL, which assumes one or a few samples arrive at a time. To effectively learn from new data while mitigating catastrophic forgetting (McCloskey & Cohen, 1989) of previously learned knowledge, various CL methods have been proposed, including replay-based approaches (Bang et al., 2021; Seo et al., 2024), network expansion methods (Wu et al., 2022; Zhou et al., 2023), and distillation-based methods (Koh et al., 2023; Wang et al., 2024).

For the practicality of online CL, most online CL methods impose resource restrictions, such as the single training epoch and limited replay memory, which restrict the number of streamed samples stored (Koh et al., 2022; Wang et al., 2022a). While the 'one-epoch training' may give a rough sense of the computational constraint, the actual budget varies across methods (Prabhu et al., 2023; Ghunaim et al., 2023) since each method requires a different amount of computations in a single epoch. Several rehearsal-based CL methods require additional storage to store the previous models and logits (Buzzega et al., 2020; Zhou et al., 2023), which was usually not included in the memory budget, which mainly considers the size of episodic memory to store samples in previous tasks. To this end, we compare CL methods with the same computational and memory budget considering all storage and computational costs. We argue that the *total budget* of memory and computation will ensure the practicality of the proposed online CL algorithms.

For a fair comparison in computational budget, we use training FLOPs per sample instead of the number of epochs, as some methods require significantly more computations per epoch than others.

---

[*]Equal contribution. [†] is affiliated with ECE, IPAI & ASRI and is a corresponding author.

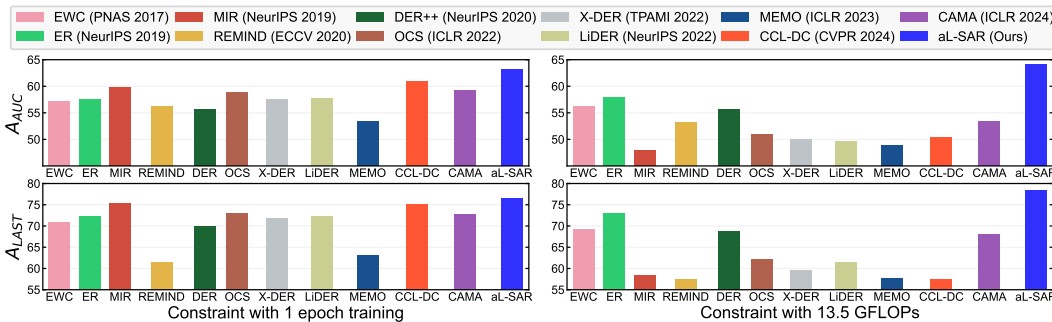

Figure 1: **Comparison of CL methods w/o total constraint (left) and w/ total constraint (right) on CIFAR-10 Gaussian setup.** In the left plot, we compare CL methods with the same number of iterations and the same episodic memory size, *i.e.*, conventional setup. In the right plot, we compare CL methods with the same training FLOPs and a fixed storage budget that includes both episodic memory and model storage, *i.e.*, our total-constrained CL setup. Compared to the conventional setup, aL-SAR shows improved performance under the total-constrained setup, since it can utilize the saved computational cost for further training. $A_{\text{AUC}}$ and $A_{\text{LAST}}$ refer to the area under the curve of accuracy and last accuracy (*i.e.*, accuracy at the end of CL), respectively.

FLOPs provide an exact measure of computational budget regardless of implementation details (Korthikanti et al., 2023), following Zhao et al. (2023); Ghunaim et al. (2023). For the same memory budget, we use an aggregated budget for various forms of extra storage, including replay memory, logits, and model parameters, converting them into bytes to obtain the actual memory cost, following Zhou et al. (2023). Upon comparing the results under fair cost, we found that the performances were often different from what was reported in the original papers proposing each method.

Considering the total memory and computational budget, we propose an online CL method using a computation-aware layer freezing strategy. Specifically, we propose to *selectively* learn (or freeze) layers per a mini-batch based on the previously learned information to reduce computation In particular, as more layers are frozen, training computation can be saved at the cost of losing information for the mini-batch since frozen layers cannot acquire any information. Considering this trade-off, we propose 'adaptive layer freezing', which chooses the best layers to freeze by maximizing the Fisher Information (FI) gained by the model for each batch, given a fixed computation budget. Unlike previous freezing methods (Lee et al., 2019; Hayes et al., 2020; Yuan et al., 2022), which predefine the rule of when and which layers to freeze, not generalizing to various datasets and models, we adaptively select layers to freeze based on the varying information in each batch.

For the loss of accuracy by the reduced computational cost, we propose a novel sample retrieval method to improve accuracy with negligible training cost. While several online CL methods, such as MIR (Aljundi et al., 2019a) and ASER (Shim et al., 2021), aim to retrieve more informative training batches than random retrieval from replay memory, they require forward/backward passes of a large candidate set to select informative batches, increasing computation by 4 times and 3times compared to ER (Rolnick et al., 2019), respectively. Applying these methods negates the computational savings of the proposed adaptive layer freezing approach. To this end, we propose to retrieve samples that the model has not learned much about information stored in episodic memory for training without incurring additional computational costs. To quantify the degree of learning, we employ the frequency of recent usage of each sample in training and the similarity of the gradients between classes. They are acquired during the training process, *i.e.*, no need for additional inference.

For our empirical validations, we compare the state-of-the-art online CL algorithms under the same FLOPs of computations and the same bytes of storage in Fig. 1. We observe that several high-performance CL methods do not maintain competitiveness under fixed FLOPs and memory budget, unexpectedly trailing behind a simple Experience Replay(Rolnick et al., 2019). On the contrary, the proposed method outperforms them by a noticeable margin under the same total budget.

We summarize our contributions as follows:

- Proposing to measure computational and memory budgets of CL algorithms by using training FLOPs and total memory size in Bytes, to fairly compare different algorithms.
- Proposing a computationally efficient adaptive layer freezing that maximizes FI per computation, as well as a memory retrieval strategy that prioritizes samples that the model has learned least.
- Empirical analysis on the computational and memory costs of various CL algorithms, showing that many state-of-the-art CL methods are less beneficial under the same budget and showing that the proposed method outperforms them by a noticeable margin across multiple benchmarks.

## 2 RELATED WORK

### 2.1 ONLINE CONTINUAL LEARNING WITH MEMORY BUDGET

Replay-based online CL methods use episodic memory and consider the memory budget. Since we also consider using episodic memory, we review them in detail as follows. Replay-based methods (Aljundi et al., 2019b; Bang et al., 2021; Koh et al., 2022) store part of the past data stream in episodic memory to replay them in future learning. Although there are simple sampling strategies such as random sampling (Guo et al., 2020) and reservoir sampling (Vitter, 1985), they are often insufficient to adapt to changing data distributions. Rather than simple methods, researchers have developed advanced sampling strategies considering factors such as uncertainty (Bang et al., 2021), importance (Koh et al., 2022), and gradient (Tiwari et al., 2022). However, these advanced methods often entail a high computational overhead, making them impractical for deployment in real-world applications. RM (Bang et al., 2021) requires a significant amount of computational cost to calculate the uncertainty for diversified sampling. Similarly, CLIB (Koh et al., 2022) involves additional forward and backward passes to calculate the decrease in loss for each batch iteration.

In addition to the memory management schemes, researchers investigate the memory usage schemes, *i.e.*, sample retrieval strategies from the rehearsal buffers. In addition to random retrieval (Chaudhry et al., 2019), the determination of retrieval based on the degree of interference (Aljundi et al., 2019a) and the adversarial Shapley value (Shim et al., 2021) has been explored. However, such methods require an inference of candidate samples, which leads to a nontrivial amount of computation in computing the loss (Aljundi et al., 2019a) or the Shapely value (Shim et al., 2021).

### 2.2 LAYER FREEZING

Freezing layers have been investigated to reduce computational costs during training in joint training (*i.e.*, ordinary training scenario other than CL) (Brock et al., 2017; Xiao et al., 2019; Goutam et al., 2020). A common freezing approach (Wang et al., 2023; Li et al., 2022) includes determining whether to freeze a layer, based on the reference model and representation similarity, such as CKA (Cortes et al., 2012) and SP loss (Tung & Mori, 2019). Additionally, EGERIA (Wang et al., 2023) unfreezes layers based on changes in the learning rate.

However, in CL, both online and offline, it is challenging to determine when to freeze a layer because metrics such as Euclidean distance and CKA cannot be used to compare the degree of convergence compared to the reference model (Mirzadeh et al., 2020). Additionally, continual learning involves a non-*i.i.d.* setup, where the data distribution continues to change (Criado et al., 2022). Therefore, in addition to changes in learning rate, it is important to consider the current data distribution when determining whether to freeze or unfreeze a layer in continual learning. (Hayes et al., 2020) have explored freezing methods for continual learning. However, they use predefined freezing configurations such as the freezing backbone block 0 after task 1, while our freezing method adaptively freezes the layers using information per batch.

## 3 APPROACH

For efficient learning in computation and storage budget, we consider two strategies; (1) reducing the computational cost of each iteration and (2) reducing the number of iterations. To implement both strategies, we propose a method employing two techniques; (1) adaptive layer-freezing and (2) similarity-aware retrieval of samples from episodic memory.

Specifically, for every training batch, the adaptive layer freezing method adaptively freezes layers so that the amount of information that can be gained from the mini-batch is maximized relative to the required computation. The memory retrieval method retrieves training batches that the model has not learned sufficiently using the number of times each sample has been used for training, *i.e.*, use-frequency, and class-wise gradient similarity. This allows the model to learn the same amount of knowledge as using random retrieval in fewer iterations, consequently reducing the overall number of training iterations. We call our method **adaptive Layer freezing and Similarity-Aware Retrieval (aL-SAR)**, illustrating the gradient update procedure of the proposed aL-SAR in Fig. 2 and providing a pseudocode in Sec. A.3.

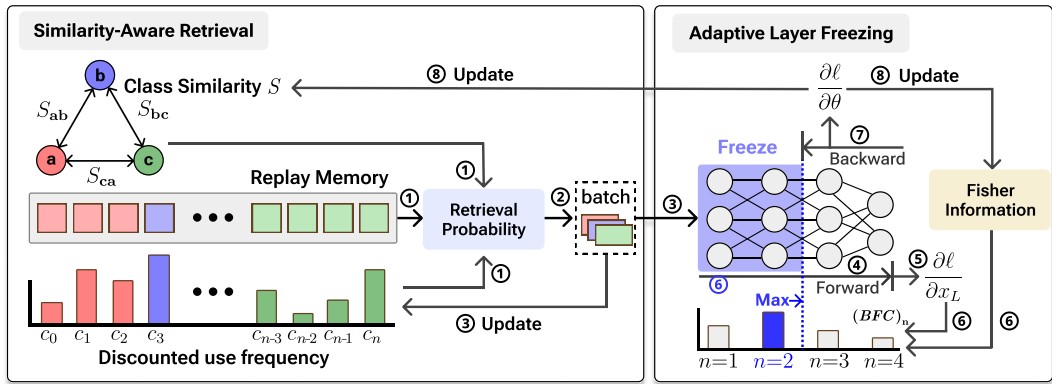

Figure 2: **Gradient update procedure of the proposed aL-SAR.** The colors in the 'Similarity-Aware Retrieval' box denote different classes. (1) 'Retrieval Probability' is calculated using class similarity $S$ and discounted use frequency $c_i$, where $c_i$ tracks the number of times the $i^{\text{th}}$ sample has been used for training. (2) A batch is sampled from memory by the 'Retrieval Probability' and (3) $c_i$ is updated by retrieval results. After the (4) forward pass of the model with the batch, (5) we compute the freezing criterion $(\text{BFC})_n$ for each layer $n$ of the model, using Fisher Information (FI) and $\frac{\partial \ell}{\partial x_L}$ (Sec. 3.1). (6) Layers 1 to $n_{\max} = \arg\max_n (\text{BFC})_n$ (in this example, $n_{\max} = 2$) are frozen in the (7) backward pass. (8) $S_{ij}$ and FI are updated using the gradient $\frac{\partial \ell}{\partial \theta}$ obtained from the backward pass.

## 3.1 ADAPTIVE LAYER FREEZING FOR ONLINE CL

To reduce the computational cost of learning a neural network with minimal accuracy loss, there have been several studies on the freezing of neural network layers in non-CL literature (Liu et al., 2021; He et al., 2021; Wang et al., 2023). These methods often rely on the learning convergence of each layer to determine which layers to freeze since converged layers no longer require further training. However, in online CL, achieving convergence is often challenging due to the limited training budget and the ever-evolving training data distribution. This requires a new approach to determine when and which layers to freeze for the incoming data in the online CL scenarios.

**Selectively Freezing Layers by Maximum Fisher Information (FI).** For a computationally efficient freezing criterion in online CL, we propose to freeze layers that learn little information per computational cost by measuring 'information' ($I$) gained by each layer during training. Here, we define the information ($I$) using Fisher Information (FI), since it is a widely used metric to measure the amount of information that each parameter in a neural network obtains from data (Durant et al., 2021; Desjardins et al., 2015; Ollivier, 2015). So, we use FI to measure the layer-wise information that each layer gains from the input data to determine which layers to freeze. Note that freezing some layers facilitates training a model for more mini-batches within a fixed computational budget as it reduces the computations per mini-batch.

However, as a trade-off, as the number of frozen layers increases, the number of updated parameters decreases, reducing the amount of information obtained per mini-batch. To maximize the information ($I$) in the model while minimizing the computational cost ($C$), we propose to maximize the expected amount of information gained per computation ($I/C$).

Formally, we try to find $n_{\max} = \arg\max_n (I/C)_n$ for $n \in [1, L]$ where we define $(I/C)_n$ as the amount of information gained per computational cost when updating the model with layers 1 to $n$ frozen. $L$ refers to the total number of layers. To compute $(I/C)_n$, we factorize it as:

$$(I/C)_n = (I/\text{mb})_n \cdot (\text{mb}/C)_n, \tag{1}$$

where 'mb' refers to the mini-batch. $(I/\text{mb})_n$ and $(\text{mb}/C)_n$ represent the amount of information gained per mini-batch and the number of mini-batch iterations per computation, respectively, when layers 1 to $n$ are frozen.

**Amount of Information Gained per Mini-batch $(I/\text{mb})_n$.** To compute $(I/\text{mb})_n$, we use $F(\theta_i)$, Fisher Information Matrix $F(\theta)$ of layer $i$, where $\theta$ and $\theta_i$ denote the parameters of the model $p_\theta(\cdot)$ and the parameters of the layer $i$ of $p_\theta(\cdot)$, respectively. But computing all components of $F(\theta_i)$ is costly as Hessian is required, which involves second-order derivatives and can be computationally inefficient. To avoid the cost, we use first-order approximation of $F(\theta_i)$ by using the diagonal components of the $F(\theta_i)$ following (Kirkpatrick et al., 2017; Soen & Sun, 2021), *i.e.*, using the

trace operator $\text{tr}(\cdot)$ as:

$$(I/\text{mb})_n = \sum_{i=n+1}^{L} \text{tr}(F(\theta_i)), \text{ where } F(\theta_i) = \mathbb{E}_{p_\theta(z),\, z \in \mathcal{D}} \left[ \left( \frac{\partial \ell}{\partial \theta_i} \right) \cdot \left( \frac{\partial \ell}{\partial \theta_i} \right)^\mathsf{T} \right], \qquad (2)$$

where $\mathcal{D}$ is the data stream, $z \in \mathcal{D}$ is the training data batch in the data stream, and $\ell = \log p_\theta(z)$ is the loss function. Note that since layers 1 to $n$ are frozen, the model gains information of layers $n+1$ to $L$, the unfrozen layers only.

**Number of Mini-batch per Computational Cost $(\text{mb}/C)_n$.** To compute $(\text{mb}/C)_n$, we initially determine its inverse, denoted $(C/\text{mb})_n$, which represents the computational cost per mini-batch when layers 1 to $n$ are frozen. We calculate $(C/\text{mb})_n$ by splitting it into forward and backward propagation as: $(C/\text{mb})_n = \sum_{i=1}^{L} (\text{FF})_i + \sum_{i=n+1}^{L} (\text{BF})_i$, where $(\text{FF})_i$ and $(\text{BF})_i$ refer to the forward FLOPs and the backward FLOPs of layer $i$, respectively. By taking an inverse of it, we obtain

$$(\text{mb}/C)_n = \frac{1}{\sum_{i=1}^{L} (\text{FF})_i + \sum_{i=n+1}^{L} (\text{BF})_i}. \qquad (3)$$

Note that since layers 1 to $n$ are frozen, no backward computation is performed for those layers. When the number of frozen layers (*i.e.*, $n$) increases, the number of layers performing backward operations is reduced, *i.e.*, $\sum_{i=n+1}^{L} (\text{BF})_i$ decreases, thus leading to an increase in the possible number of mini-batch within the given computational budget $C$.

Combining Equation 2 and Equation 3, we can finally calculate $(I/C)_n$, which represents the information the model can gain within a given computational budget when layers 1 to $n$ are frozen, by a product of $(I/\text{mb})_n$ and $(\text{mb}/C)_n$:

$$(I/C)_n = (I/\text{mb})_n \cdot (\text{mb}/C)_n = \frac{\sum_{i=n+1}^{L} \text{tr}(F(\theta_i))}{\sum_{i=1}^{L} (\text{FF})_i + \sum_{i=n+1}^{L} (\text{BF})_i}. \qquad (4)$$

By freezing layer 1 to layer $n_{\max} = \arg\max_n (I/C)_n$, we can maximize the expected amount of information gained per computation during training.

**Batch-wise Freezing for Online CL.** $(I/C)_n$ is supposed to be calculated on the entire data stream since $F(\theta)$ is defined as an expectation over the whole dataset in Equation 2. Thus, $(I/C)_n$ does not account for the variable amount of information of each batch. In online CL where the incoming batch distribution continuously shifts, the variation is not negligible. Specifically, for batches containing sufficiently trained classes, it is advantageous to freeze more layers, while for batches with insufficiently trained classes, it is beneficial to freeze fewer layers. To address this, we propose the batch freezing criterion (BFC) which quantifies the *net benefit* in the information gained by the model when we freeze layers given an input batch $z_t$.

To define the BFC, we compare (1) the amount of information we would lose from the current batch by freezing and (2) the expected amount of information we would gain in the future using the saved computation from freezing. Then, we can estimate the net benefit from freezing in terms of the information gained by the model by subtracting (1) from (2).

To estimate (1), we use the trace of layer-wise FI, *i.e.*, $\text{tr}(F(\theta_i))$, defined in Equation 2. Note that $F(\theta_i)$ is the FI over the whole dataset; thus we convert it into batch-specific FI $F_{z_t}(\theta_i) = \mathbb{E}_{p_\theta(z),\, z \in z_t} \left[ \left( \frac{\partial \ell}{\partial \theta_i} \right) \cdot \left( \frac{\partial \ell}{\partial \theta_i} \right)^\mathsf{T} \right]$ by multiplying the ratio of the FI of the batch relative to the FI of the entire dataset. Using $F_{z_t}(\theta_i)$, the amount of information we would lose from the current batch $z_t$ by freezing layers 1 to $n$ is obtained as $\sum_{i=1}^{n} \text{tr}(F_{z_t}(\theta_i))$. Please refer to Sec. A.1 for more details on the estimation of $F_{z_t}(\theta_i)$ from $F(\theta_i)$.

To estimate (2), we calculate the amount of computations saved by freezing layers 1 to $n$, and the amount of information we can obtain from the saved computations. Here, the saved computation refers to the sum of the backward FLOPs of the frozen layers, denoted as $\sum_{i=1}^{n} (\text{BF})_i$. Using $(I/C)_n$ defined in Equation 4, we estimate the expected information $(I)$ obtainable from the saved computations $(C)$ by multiplying the saved computation and $(I/C)_n$. With optimal freezing that maximizes $(I/C)$, the anticipated information obtained from the saved calculations is $\max_m (I/C)_m \cdot \sum_{i=1}^{n} (\text{BF})_i$.

Finally, by subtracting (1) from (2), we obtain $\text{BFC}(z_t)_n$ as:

$$\text{BFC}(z_t)_n = \max_m (I/C)_m \cdot \sum_{i=1}^{n} (\text{BF})_i - \sum_{i=1}^{n} \text{tr}\left(F_{z_t}(\theta_i)\right). \tag{5}$$

The positive $\text{BFC}(z_t)_n$ implies that freezing layer 1 to $n$ is beneficial in terms of information, and a negative value indicates otherwise. By freezing layer 1 to $n_{\max}(z_t) = \arg\max_n \text{BFC}(z_t)_n$, we can select the most beneficial freezing strategy for batch $z_t$. We argue that it allows us to dynamically select the freezing strategy by the learning capacity of the layers and the batch's informativeness. We empirically demonstrate the distribution of the $\text{BFC}(z_t)_n$ in Sec. A.10.

## 3.2 SIMILARITY-AWARE RETRIEVAL BASED ON 'USE-FREQUENCY'

In rehearsal-based CL methods, sample retrieval strategies such as MIR (Aljundi et al., 2019a) and ASER (Shim et al., 2021) have a high computational cost, requiring multiple additional model inferences. Thus, in the same computational budget, their performances are surprisingly inferior to simple random retrieval (*i.e.*, ER) as shown in Fig. 1. Here, we propose a computationally efficient sample retrieval strategy that does not require additional model inference.

In online CL, new data continuously stream in, and old data remain in memory, causing an imbalance in 'the number of times each sample is used for training', which we call '*use-frequency*.' We argue that samples with high use-frequency yield marginal knowledge gains in training, and samples with low use-frequency are likely to contain knowledge that the model has not yet learned. So, we propose to sample data with low use-frequency for training with high probability.

**Discounted Use Frequency ($c_i$).** But using the use-frequency for sampling does not consider the knowledge forgetting in the CL setup. If a sample was frequently used in the past but seldom used in recent iterations, its knowledge may have been forgotten, despite its high use-frequency. Inspired by the exponential decaying model of forgetting (Shin & Lee, 2020; Mahto et al., 2020; Chien et al., 2021), we propose a decay factor ($0 < r < 1$) in the use frequency at each iteration, calling *'discounted-use-frequency'* for $i^{\text{th}}$ sample ($c_i$). For example, if $i^{\text{th}}$ sample is used $n$ times for training at a specific time point, after $t$ iterations, we define its discounted use frequency as $c_i = n \cdot r^t$.

**Effective Use Frequency ($\hat{c}_i$).** However, the model can learn knowledge about a sample by training other similar samples (*e.g.*, samples from the same classes). Thus, $c$ of the other samples could also affects $c_i$ of the sample. These similar samples *effectively* increase the use-frequency of the particular sample. At the same time, the model may lose knowledge about the sample when training on different samples (*e.g.*, from other classes), effectively decreasing the use-frequency.

To account for this, we define *'effective-use-frequency'* by adding the other samples' use-frequency multiplied by the similarity of samples to $c_i$. For the sample similarity score, inspired by (Du et al., 2018), which uses gradient similarity to assess the helpfulness or harmfulness of an auxiliary task to the original task, we hypothesize that samples with similar gradients bear similar information. So, for the proxy of sample similarity, we use cosine similarity between the gradients.

However, tracking the gradient similarities between all sample pairs requires excessive memory ($\sim 10^{12}$ pairs for ImageNet) and computation. Thus, we approximate it to class-wise similarities, which is the expected gradient similarity between samples from two classes. Formally, we define the class-wise similarity $\mathcal{S}_{y_1, y_2}$ for classes $y_1$ and $y_2$ as:

$$\mathcal{S}_{y_1, y_2} = \mathbb{E}_{z_1 \in D_{y_1}, z_2 \in D_{y_2}} \left[\cos(\nabla_\theta l(z_1), \nabla_\theta l(z_2))\right], \tag{6}$$

where $D_{y_i}$ is the training data for class $y_i$ and $\nabla_\theta l(z_i)$ is gradient of $i^{\text{th}}$ sample. Using the approximated class-wise similarities, we define the effective-use-frequency $\hat{c}_i$ for the $i^{\text{th}}$ sample as:

$$\hat{c}_i = c_i + \sum_{y \in \mathcal{Y}} \mathcal{S}_{y, y_i} \cdot C_y, \tag{7}$$

where $\mathcal{Y}$ is the set of all seen classes, $\mathcal{S}_{y, y_i}$ is a class similarity between class $y$ and $y_i$, and $C_y = \sum_{y_j = y} c_j$ is the sum of the discounted use-frequencies for all samples of class $y$.

Unfortunately, calculating the expected value in $\mathcal{S}_{y_i, y_j}$ (Equation 6) from scratch for each iteration requires a gradient calculation for all samples in the classes $y_i$ and $y_j$, which is computationally

expensive. As a computationally efficient alternative, we further propose using the EMA to update the previous estimate of $S_{y_i,y_j}$ rather than calculating the expectation from scratch. Note that we only utilize gradients from unfrozen layers, obtained during the training process, *i.e.*, incurring no additional cost. Please refer to Sec. A.18 for detailed information on the calculation of $\mathcal{S}_{y_i,y_j}$.

Finally, we obtain the retrieval probabilities $p_i$ for $i^{\text{th}}$ sample with the effective-use-frequency as:

$$p_i = \frac{e^{-\hat{c}_i/T}}{\sum_{j=1}^{|\mathcal{M}|} e^{-\hat{c}_j/T}}, \tag{8}$$

where $T$ is a temperature hyperparameter. Samples with low $\hat{c}_i$ have high chances of being retrieved, so insufficiently trained samples are preferred to sufficiently trained ones, thereby accelerating training. We present several toy experiments and an ablation study on the components of SAR to validate the empirical benefit of our retrieval algorithm in Sec. A.26 and Sec. A.8, respectively.

## 4 Experiments

### 4.1 Setup

For empirical validation, we adopt the total budget for memory and computation. For the memory budget, we use Bytes following (Zhou et al., 2023), which considers memory costs not only for the samples in episodic memory but also for additional model parameters used in regularization or distillation. For the computational budget, we use training FLOPs. For the dataset, we use CIFAR-10/100, CLEAR-10/100, and ImageNet-1K. We evaluate the methods in a conventional disjoint task setup and a newly proposed Gaussian task setup (Shanahan et al., 2021; Wang et al., 2022b; Koh et al., 2023). For all experiments, we averaged 3 different random seeds, except ImageNet-1K due to computational cost (Bang et al., 2021; Koh et al., 2023). We conducted a Welch's $t$-test with a significance level of 0.05. We highlighted the highest performance in bold. In cases where statistical significance was not observed, we underlined all other results within the significance level.

**Metrics.** We report the last accuracy $A_{\text{last}}$ and the area under the curve of accuracy $A_{\text{AUC}}$ (Koh et al., 2022). The $A_{\text{last}}$ measures the accuracy at the end of CL. The $A_{\text{AUC}}$ measures the accuracy per time step using the accumulated test set of all previously seen tasks and then computes the area under the accuracy curve. For each evaluation, we evaluate using the entire test set for the classes seen so far. We argue that $A_{\text{AUC}}$ is a suitable metric to measure prompt learning of new knowledge.

**Baselines.** We compare aL-SAR to state-of-the-art online CL methods such as ER (Rolnick et al., 2019), DER++ (Buzzega et al., 2020), MIR (Aljundi et al., 2019a), MEMO (Zhou et al., 2023), RE-MIND (Hayes et al., 2020), EWC (Kirkpatrick et al., 2017), OCS (Yoon et al., 2022), LiDER (Bonicelli et al., 2022), X-DER (Boschini et al., 2023), CCL-DC (Wang et al., 2024), and CAMA (Kim et al., 2024). We describe the implementation details and hyperparameters in Sec. A.4.

### 4.2 Quantitative Analysis

We evaluated CL methods, including aL-SAR, with strictly restricted computation and memory budgets as specified in Sec. 4.1. Note that while we set the training iterations of aL-SAR to be the same as other baselines, aL-SAR adaptively freezes layers, resulting in fewer FLOPs consumed. For experiments on various computational and memory budgets, we use the relatively small CIFAR datasets to cover a wide range of given budgets. To validate our methods on large datasets and datasets with temporal domain shift, we also show experiments on ImageNet and CLEAR datasets. Note that these experiments are conducted on various CL setups, including Gaussian task setup, Disjoint task setup, and domain-incremental setup. Furthermore, we applied our adaptive layer freezing method to LLaVA-v1.5-7B, demonstrating a significant reduction in the computational cost of training the multi-modal large language model while preserving performance.

**Various Computational Budget under the Same Memory Budget.** We compare CL methods under fixed memory budgets and various computational budgets in Fig. 3. We observe that aL-SAR significantly outperforms others in all datasets and both setups, especially under a low computational budget. It shows that our similarity-aware retrieval effectively promotes rapid learning by retrieving informative training batches even with a limited computational budget, while random retrieval and MIR (Aljundi et al., 2019a) require high computation to achieve comparable performance.

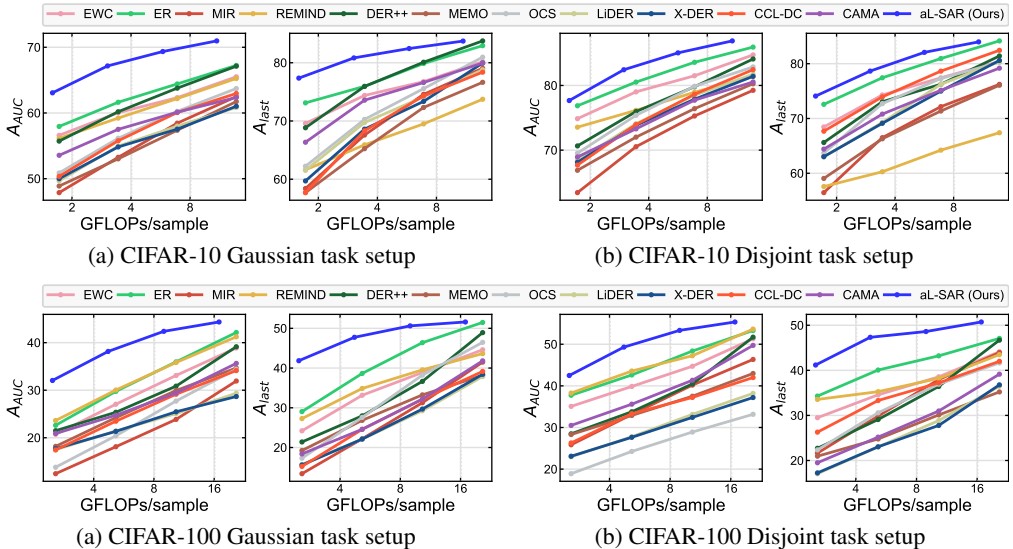

Figure 3: Accuracy on Gaussian and Disjoint CL setup in CIFAR-10 and CIFAR-100 for a wide range of FLOPs per sample. aL-SAR outperforms all CL methods compared. We use a memory budget of 30.4MB.

Furthermore, we observe a notable increase in the FLOPs saved by aL-SAR through freezing, particularly pronounced at higher computational budgets. As the model undergoes more iterations, the amount of information that the model gains from the training data decreases. Thus, our adaptive layer freezing adaptively adjusts the freezing criterion to freeze more layers, leading to lower FLOPs, thus the line stops at the earlier GFLOPs value than the baselines. We provide comprehensive analysis when varying computations under various memory constraints in Sec. A.5. Please refer to Sec. A.21 for more details on the computational budget.

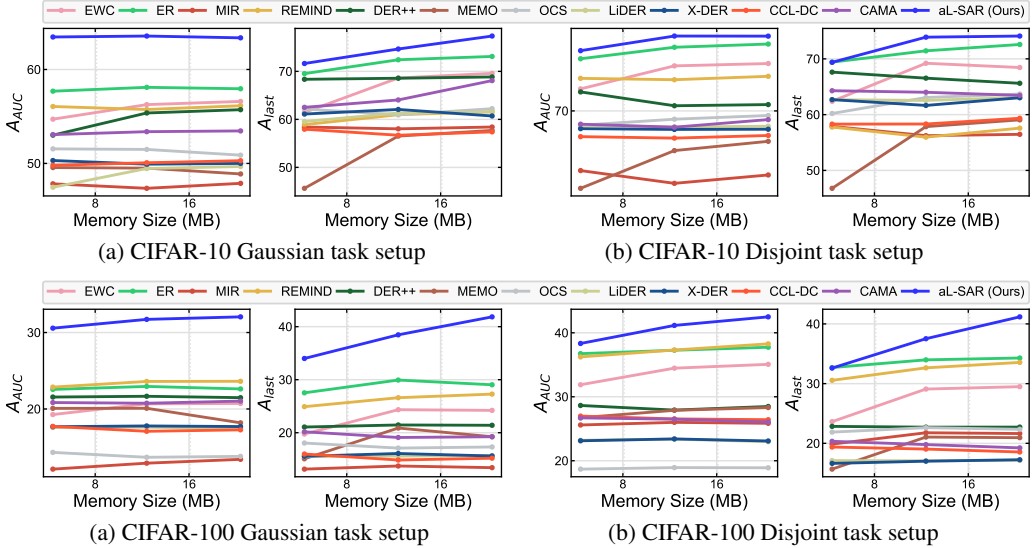

Figure 4: Accuracy on Gaussian and Disjoint CL setup in CIFAR-10 and CIFAR-100 for various memory budget. aL-SAR outperforms all CL methods compared. We use a computational budget of 171.94 TFLOPs.

**Various Memory Budget under the Same Computational Budget.** We now fix the computational budget and test various memory budgets for CL methods in Fig. 4. With its minimal additional memory usage and effective utilization of episodic memory through similarity-aware retrieval, aL-SAR again outperforms other methods by a significant margin in all datasets, indicating its suitability for both large and small memory budgets.

**Large Datasets and Temporal Domain Shifts.** To investigate the scalability of CL methods on large datasets and temporal domain shifts, we report the performances on ImageNet-1K Gaussian setup and CLEAR-10/100 with a fixed computational and memory budget in Tab. 1. In the

| Methods | ImageNet-1K | | CLEAR-10 | | CLEAR-100 | |
|---|---|---|---|---|---|---|
| | $A_{\text{AUC}}$ ↑ | $A_{\text{last}}$ ↑ | $A_{\text{AUC}}$ ↑ | $A_{\text{last}}$ ↑ | $A_{\text{AUC}}$ ↑ | $A_{\text{last}}$ ↑ |
| EWC (Kirkpatrick et al., 2017) | 20.05 | 32.55 | 62.40±0.17 | 71.25±1.29 | 29.62±0.44 | 40.36±1.34 |
| ER (Rolnick et al., 2019) | 20.01 | 32.60 | 64.00±0.06 | 72.28±1.15 | 31.56±0.42 | 44.90±0.74 |
| ER-MIR (Aljundi et al., 2019a) | 7.45 | 17.04 | 62.29±0.17 | 70.52±1.13 | 23.24±0.14 | 34.92±1.16 |
| REMIND (Hayes et al., 2020) | 15.58 | 22.42 | 62.64±0.04 | 70.68±0.90 | 26.76±0.30 | 30.99±0.55 |
| DER++ (Buzzega et al., 2020) | 23.51 | 29.82 | 61.30±0.77 | 71.52±1.02 | 25.83±0.43 | 39.89±0.89 |
| OCS (Yoon et al., 2022) | 6.90 | 19.57 | 58.69±0.25 | 68.69±1.21 | 15.24±1.27 | 22.52±2.02 |
| LiDER (Bonicelli et al., 2022) | 27.65 | 34.66 | 58.36±0.75 | 69.16±1.21 | 28.75±0.28 | 39.92±1.29 |
| X-DER (Boschini et al., 2023) | 27.46 | 34.47 | 58.19±0.23 | 69.70±0.97 | 23.01±0.15 | 33.70±0.70 |
| MEMO (Zhou et al., 2023) | 11.15 | 19.98 | 60.29±0.31 | 68.69±1.21 | 20.33±0.28 | 25.53±0.92 |
| CAMA (Kim et al., 2024) | 7.94 | 17.44 | 60.97±0.52 | 71.66±1.18 | 26.27±0.37 | 39.42±0.67 |
| CCL-DC (Wang et al., 2024) | 19.70 | 32.12 | 61.08±0.55 | 71.25±0.82 | 23.81±0.32 | 35.07±0.67 |
| aL-SAR (Ours) | **33.24** | **35.02** | **67.61±0.45** | **74.36±0.80** | **38.65±0.30** | **47.56±0.37** |

Table 1: **Quantitative comparison between different CL methods on CIL setup.** aL-SAR outperforms baselines with a lower computational budget and the same storage budget. Specifically, while all other baselines consume 112,000, 716, and 5,250 TFLOPs, aL-SAR consumes 93,000, 598, and 4,059 TFLOPs in the ImageNet-1K Gaussian setup, CLEAR-10, and CLEAR-100, respectively. The memory budget is fixed at 5,736MB, 564MB, and 1,148MB in ImageNet-1K, CLEAR-10, and CLEAR-100, respectively.

ImageNet-1K Gaussian setup, aL-SAR outperforms other methods with slightly fewer FLOPs. In this setup, training distribution shifts continuously, so the model has to constantly adapt to the new distribution, resulting in less freezing and more benefit from the fast adaptation enabled by the retrieval method.

We then investigate CL methods under temporal domain shift with fixed computational and memory budget using the CLEAR-10/100 datasets. Unlike the class-incremental, where new classes are added, this domain-incremental setup introduces new domains, while the classes remain the same. As shown in the table, aL-SAR also outperforms the state-of-the-art in domain-incremental setups. It is partly because our retrieval method balances the use-frequency of samples in different domains so that the model learns more on relatively less-learned domains, allowing fast adaptation to new domains. Note that aL-SAR also significantly saves FLOPs thanks to adaptive layer freezing. Please refer to Sec. A.23 and Sec. A.21 for more details about the memory budget and the computational budget, respectively. Moreover, we provide extensive experimental results in the memory infinite setup and comparisons with efficient CL baselines in Sec.A.22 and Sec.A.12, respectively.

**Comparison of aL with Layer Freezing Methods.** We compare our proposed adaptive freezing method (aL) with REMIND (Hayes et al., 2020) and PTLF (Yang et al., 2023), layer freezing methods for offline CL, as well as EGERIA (Wang et al., 2023), a layer freezing method for non-CL. The results are summarized in Tab. 2. As we can see in the table, PTLF and EGERIA show negligible differences in TFLOPs with No Freezing, indicating minimal layer freezing. While REMIND significantly reduces training FLOPs, it leads to performance degradation. In contrast, our proposed aL significantly reduces training FLOPs while maintaining performance. Notably, REMIND, PTLF, and EGERIA require task identity information during training, while aL operates without it. This further highlights the practicality and adaptability of aL in online CL setups. We provide a detailed analysis of the results and further comparisons in the disjoint setup in Sec.A.15 for the sake of space.

| Methods | CIFAR-10 | | | CIFAR-100 | | |
|---|---|---|---|---|---|---|
| | $A_{\text{AUC}}$ ↑ | $A_{\text{last}}$ ↑ | TFLOPs ↓ | $A_{\text{AUC}}$ ↑ | $A_{\text{last}}$ ↑ | TFLOPs ↓ |
| No freezing | **64.60 ± 0.83** | 72.43 ± 0.38 | 171.94 | **42.49 ± 0.75** | 50.49 ± 0.29 | 515.82 |
| REMIND | 59.25 ± 0.37 | 65.01 ± 1.31 | 151.31 (-12.0%) | 35.50 ± 0.56 | 40.00 ± 0.86 | 453.92(-12.0%) |
| PTLF | 63.82 ± 0.24 | 71.30 ± 0.91 | 164.14 (-4.5%) | 42.09 ± 0.58 | 49.92 ± 0.65 | 495.67 (-4.1%) |
| EGERIA | 63.95 ± 0.72 | 71.72 ± 0.81 | 169.25 (-1.6%) | 42.18 ± 0.63 | 50.05 ± 0.72 | 506.54 (-1.8%) |
| aL (Ours) | 64.38 ± 0.32 | **72.57 ± 0.79** | **146.80** (-14.6%) | 42.38 ± 0.76 | **50.62 ± 0.87** | **427.90** (-17.0%) |

Table 2: **Comparison between our proposed adaptive layer freezing and other freezing methods.** We compare them in the CIFAR-10 Gaussian setup and the CIFAR-100 Gaussian setup.

**Comparison of SAR with Memory Retrieval Methods.** We compare our proposed retrieval method, *i.e.*, SAR with ASER (Shim et al., 2021) and MIR (Aljundi et al., 2019a) in Sec. A.19.

**Application of aL in Training Multi-Modal Large Language Models (MLLMs)** We demonstrate the effectiveness of aL in training MLLMs by applying it to training of the LLaVA-1.5-7B model (Liu et al., 2023). Following Ye et al. (2023), we update only the pretrained projection MLP layers and LoRA adapters (Hu et al., 2021), keeping the LLM frozen for training efficiency. We summarize the result in Tab. 3. We provide implementation details and datasets (*i.e.*, Bongard-HOI (Jiang et al., 2022) and Bongard-OpenWorld (Wu et al., 2024)) of MLLM training in Sec. A.13. As shown in the table, aL saves approximately 12% in training FLOPs while maintaining performance, by adaptively freezing LoRA layers during the training process. We believe that aL can be effectively integrated in a plug-and-play manner with CL methods in large models, such as LLMs and MLLMs, to reduce training costs while achieving strong performance.

| Methods | Bongard-HOI-P/N | | | Bongard-OpenWorld-P/N | | |
|---|---|---|---|---|---|---|
| | $A_{AUC}$ ↑ | $A_{last}$ ↑ | TFLOPs ↓ | $A_{AUC}$ ↑ | $A_{last}$ ↑ | TFLOPs ↓ |
| No freezing | 68.01±0.47 | **65.39±0.69** | 1578.02 | **56.71±1.53** | 56.14±2.55 | 2959.61 |
| aL (Ours) | **68.62±1.77** | 64.59±0.77 | **1431.27** (-10.3%) | 55.34±1.55 | **57.17±2.61** | **2722.23** (-8.0%) |

Table 3: **The effect of adaptive freezing on MLLM training.** We use LLaVA-1.5-7B model with LoRA.

## 4.3 ABLATION STUDY

We ablate the model to investigate the benefit of each proposed component in CIFAR-10/100 and summarize the results in Tab. 4. In Tab. 4, similarity-aware retrieval (SAR) increases the performance while using the same number of iterations. This shows that SAR increases the amount of knowledge learned per iteration, as we claim in Sec. 3. While computational cost also increases, its increase is modest compared to other retrieval methods such as MIR (Aljundi et al., 2019a) or ASER (Shim et al., 2021) that require $3 \sim 4\times$ more computations. Furthermore, we observe that the aL significantly reduces FLOPs with negligible loss in accuracy. In summary, our method outperforms the baseline while using fewer FLOPs than the baseline, each by a noticeable margin. We provide further ablation studies of the proposed components in the Disjoint setup in Sec. A.7. Furthermore, we present an ablation study of the proposed retrieval method, *i.e.*, SAR, in Sec.A.8.

| Methods | CIFAR-10 | | | CIFAR-100 | | |
|---|---|---|---|---|---|---|
| | $A_{AUC}$ ↑ | $A_{last}$ ↑ | TFLOPs ↓ | $A_{AUC}$ ↑ | $A_{last}$ ↑ | TFLOPs ↓ |
| Vanilla | 60.76±0.11 | 70.08±0.97 | 163.74 | 31.97±0.89 | 37.80±1.30 | 245.91 |
| + aL | 60.38±0.54 | 69.04±0.83 | **142.23** | 31.77±0.60 | 38.03±0.35 | **217.40** |
| + SAR | **64.60±0.83** | 72.43±0.38 | 171.94 | **37.60±0.40** | **42.69±0.18** | 257.97 |
| + aL & SAR (Ours) | 64.38±0.32 | **72.57±0.79** | 146.80 | 37.20±0.73 | 42.55±0.79 | 221.49 |

Table 4: Benefits of the proposed components of aL-SAR, adaptive layer freezing (aL) and similarity-aware retrieval (SAR), in Gaussian task setup. 'Vanilla' is a simple replay-based method that trains on randomly retrieved batches from a balanced reservoir memory. The memory budget is 7.6MB for CIFAR-10 and 13.44MB for CIFAR-100. We train for 1 iter per sample for CIFAR-10 and 1.5 iter per sample for CIFAR-100.

Moreover, we provide detailed studies in the appendix for the space sake. Specifically, we investigate the performance with different freezing strategies in Sec. A.6, the detailed effect of freezing on accuracy and FLOPs in Sec. A.9, the effect of temperature $T$ in Sec. A.20, and the application of our freezing method on Vision Transformer (ViT) (Dosovitskiy et al., 2020)) is discussed in Sec. A.16.

## 5 CONCLUSION

We address the challenge of achieving high performance on both old and new data with minimal computational cost and limited storage budget in online CL. While CL with fixed episodic memory size has been extensively studied, we have investigated the total storage budget required for online CL as well as the computational budget for developing practically useful online CL methods.

To this end, we proposed aL-SAR, a computationally efficient CL method comprising two components: similarity-aware retrieval and adaptive layer freezing. Our empirical validations show that several high-performing CL methods are not competitive under a fixed computational budget, falling behind a simple baseline of training on randomly retrieved batches from memory.

## ETHICS STATEMENT

We propose a better learning scheme for online continual learning for realistic learning scenarios. While the authors do not explicitly aim for this, the increasing adoption of deep learning models in real-world contexts with streaming data could potentially raise concerns such as inadvertently introducing biases or discrimination. We note that we are committed to implementing all feasible precautions to avert such consequences, as they are unequivocally contrary to our intentions.

## REPRODUCIBILITY STATEMENT

To further facilitate the reproduction, we provide open-source implementations of our proposed method (Sec.3), along with data splits and baseline models used in our experiments (Sec.4), available at https://github.com/snumprlab/budgeted-cl.

## ACKNOWLEDGMENT

This work was partly supported by AI Center, Samsung Electronics, and the IITP grants (No.RS-2022-II220077, No.RS-2022-II220113, No.RS-2022-II220959, No.RS-2021-II211343 (SNU AI), No.RS-2021-II212068 (AI Innov. Hub)) funded by the Korea government (MSIT).

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

## A  APPENDIX

### A.1  DETAILS ON THE ESTIMATION OF BATCH-WISE FISHER INFORMATION $F_{z_t}(\theta_i)$

To estimate the batch-wise FI $F_{z_t}(\theta_i)$ from FI $F(\theta_i)$, we compute the FI ratio, *i.e.*, the ratio of the FI of the batch $z_t$ to the FI of the entire dataset, using the property that the FI is quadratically proportional to the magnitude of the gradient (equation 2). To be specific, we compute the ratio between $|\nabla_x \ell(z_t)|^2$ of the current batch and the expectation of $|\nabla_x \ell(z)|^2$ on the whole dataset.

Note that, we only use the gradient of the last layer feature $x_L$ to estimate the magnitude of the gradient following (Koh et al., 2023), since the gradients of the preceding layers are proportional to the gradient of the final layer due to the chain rule. Using the estimate, we compute the batch-wise FI $F_{z_t}(\theta_i)$ from batch $z_t$ when freezing layers 1 to $n$ as:

$$F_{z_t}(\theta_i) = \frac{|\nabla_{x_L} \ell(z_t)|^2}{\mathbb{E}_z \left[|\nabla_{x_L} \ell(z)|^2\right]} \cdot F(\theta_i), \tag{9}$$

where $\mathbb{E}_z$ is the expectation over all input batches.

For the calculation of $F_{z_t}(\theta_i)$, we have to compute the expected values in the average gradient magnitude ($\mathbb{E}_z \left[|\nabla_{x_L} \ell(z)|^2\right]$) and FI ($F(\theta_i)$'s) (equation 2). Since calculating the expected values (using all samples in replay memory) in every learning iteration is computationally expensive, we estimate them by the exponential moving average (EMA) of the estimated expectations computed by the mini-batch of the past iterations. However, the EMA estimate of $\mathrm{tr}(F(\theta_i))$ requires a gradient calculation for all layers, so it cannot be used with freezing, which stops the gradient computations. Since the estimation of $\mathrm{tr}(F(\theta_i))$ and the freezing cannot be performed at the same time, at each $m$ iteration, we train (*i.e.*, unfreeze) all layers to update the estimate of $\mathrm{tr}(F(\theta_i))$ for all $i$. For the other $m-1$ iterations, we do not update $\mathrm{tr}(F(\theta_i))$ and freeze the model based on the values of (BFC), using the previously estimated value of $\mathrm{tr}(F(\theta_i))$.

### A.2  DETAILS ABOUT ESTIMATION OF FISHER INFORMATION TRACE

To check how accurate our Fisher Information trace estimate is, we ran an experiment comparing the Fisher Information trace estimated with i) a batch size of 16 once every four steps and ii) a batch size of 64 for every step (*i.e.*, 16 times larger sample size) on CIFAR-10 Gaussian task setup. We use ResNet-32 as the backbone and show the trace of the Fisher Information of the last layers for each block, *i.e.* layers 8, 16, 24, and 32. From the result in Fig. 5, we observe that the estimation with i) a batch size of 16 once every four steps does not deviate much from the estimation with ii) a batch size of 64 for every step, showing that our estimation is reasonably accurate.

### A.3  DETAILED ALGORITHM OF ALL-SAR

Algorithm 1 provides a comprehensive pseudocode for the aL-SAR method. aL-SAR has two components: similarity-aware retrieval and adaptive layer freezing. In the algorithm box, lines 3, 6-13, and 25-26 describe the similarity-aware retrieval method, and lines 15-24 describe the adaptive layer freezing method.

### A.4  IMPLEMENTATION DETAILS

We use ResNet-32 (He et al., 2016) for CIFAR-10, CIFAR-100, CLEAR-10 and CLEAR-100, and use ResNet-18 as the network architecture for ImageNet-1K. We set the training hyperparameters as follows (Prabhu et al., 2020; Bang et al., 2021; Koh et al., 2022). For CIFAR-10, CIFAR-100, CLEAR-10, and ImageNet, we use batchsize of 16, 16, 16, and 256, respectively, and Adam optimizer with LR of 0.0003 for all datasets and setup. To calculate $A_{\mathrm{AUC}}$, we use an evaluation period of 100 samples for CIFAR-10/100 and CLEAR-10/100, and 8000 samples for ImageNet-1K. For memory constraints, we used memory size of 7.6MB, 13.44MB, 25.12MB for CIFAR-10 and CIFAR-100, 617MB for CLEAR-10, 5.8GB for ImageNet.

For data augmentation, we apply RandAugment (Cubuk et al., 2020) to all CL methods. For hyperparameters, we set all the EMA ratios required for aL-SAR to 0.01 for all datasets. For the values of $k$ and $T$ used in memory retrieval, we use $k = 4$ and $T = 0.125$ for all experiments.

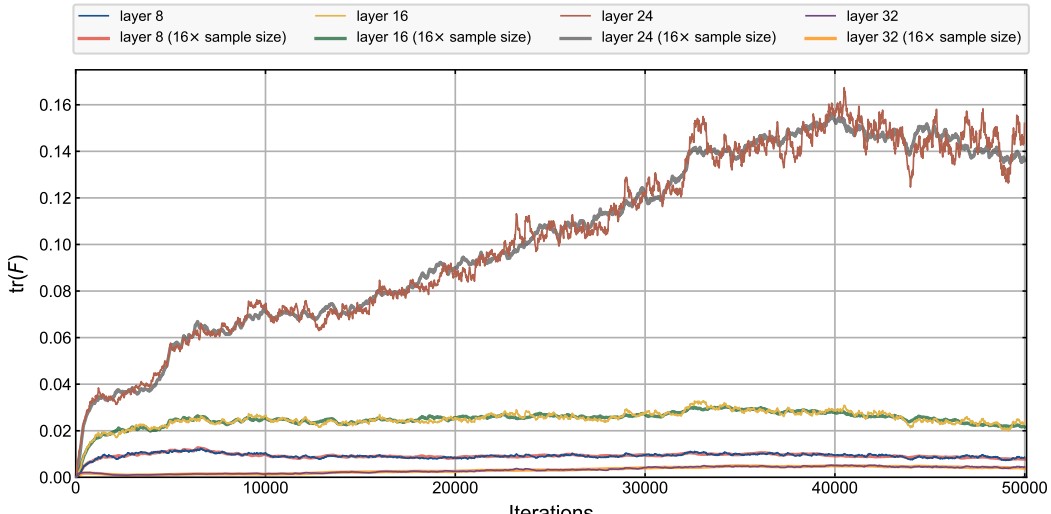

Figure 5: The estimated trace of Fisher Information for layers 8, 16, 24, and 32 of ResNet-32 on CIFAR-10 Gaussian Task setup, comparing the estimation used in aL-SAR and the estimation with a 16 times bigger sample size.

We found the hyperparameters through a search on CIFAR-10 and applied them to other datasets and setups without further tuning. This decision was made due to the lack of access to the dataset before actual training in real-world continual learning applications, as data arrive as an online stream Thus, we cannot perform a dataset-specific hyperparameter search. Similarly, we do not know the distribution of the data in the future, so a setup-specific (*e.g.*, Disjoint/Gaussian task) hyperparameter search is also not possible. Therefore, a realistic scenario is performing a hyperparameter search using a known dataset and setup in a test environment, and applying it to real-world with unknown datasets and distributions. To validate our method's ability to adapt to unknown data in this scenario, we search for hyperparameters in CIFAR-10 Gaussian task setup and apply them to all other datasets and setups.

For aL-SAR, we use memory-only training, where the training batch is retrieved from the episodic memory at every iteration. And we use the Greedy Balanced Sampling strategy (Prabhu et al., 2020) for memory sampling. We use $m = 4$ for all datasets and setups, where $m$ refers to the batch cycles where layer freezing is not applied.

For LiDER (Bonicelli et al., 2022), it follows a plug-in approach, integrated with existing methods. Based on the experimental results in the paper, the combination with X-DER showed the most promising performance. Consequently, the results obtained by combining X-DER were reported as the results for LiDER.

## A.5 EXPERIMENT RESULTS USING ADDITIONAL MEMORY CONSTRAINTS

The results obtained using 13.46MB memory budgets in CIFAR-10 and CIFAR-100 are shown in Fig. 6. In addition to results in the 7.6MB memory budget in Fig. 3, our method outperforms other methods in all tested memory budgets, further showing that our method is robust across various memory constraints.

## A.6 COMPARISON BETWEEN ADAPTIVE LAYER FREEZING AND NAIVE LAYER FREEZING

We compare the proposed adaptive layer freezing method with various naive freezing methods, in both Gaussian and disjoint setup in CIFAR-10. The results are summarized in Tab.5. Each freezing strategy chooses the number of frozen layers $n \in [0, L]$ where $L$ is the total number of layers, so that when $n \geq 1$, layer 1 to layer $n$ are frozen. The compared freezing strategies are: random freezing ($n$ is randomly selected from $[0, n_{\max}]$ every iteration for a fixed $n_max \in [0, L]$), constant freezing ($n$ is fixed initially) and linear freezing ($n$ increases linearly from 0 to $n_{\max}$ for a fixed $n_max \in [0, L]$).

---

**Algorithm 1** adaptive Layer freezing and Similarity-Aware Retrieval (aL-SAR)

---

1: **Input** model $f_\theta$, Layer parameters $\theta_l$, Training data stream $\mathcal{D}$, Batch size $B$, Learning rate $\mu$, EMA ratio $\alpha$, Frequency scale $k$, Retrieval temperature $T$, Number of layers $L$, Total Forward FLOPs (FF), Backward FLOPs per layer $(\text{BF})_l$

2: **Initialize** Episodic memory $\mathcal{M} \leftarrow \{\}$, Sample frequency $c_i \leftarrow 0$, Class frequency $C_y \leftarrow 0$, Class Similarity $S_{y_1 y_2} \leftarrow 0$, Layer Fisher trace $(trF)_l \leftarrow 0$, Expected gradient norm $|\bar{g}_{x'_L}| \leftarrow 0$

3: $\theta_S = \text{RandomSubset}(\theta, 0.0005)$
                       ▷ Random subset of $\theta$ containing $0.05\%$ of the parameters, for updating class similarity $S$

4: **for** $(x_t, y_t) \in \mathcal{D}$ **do**                                        ▷ samples from data stream

5:      **Update** $\mathcal{M} \leftarrow \text{GreedyBalancingSampler}\,(\mathcal{M} \cup (x_t, y_t))$
                                   ▷ Memory update with Greedy Balancing Sampler

6:      $\hat{c}_i = c_i + \sum_{y \in \mathcal{Y}} S_{y y_i} C_y \quad \forall\, (x_i, y_i) \in \mathcal{M}$
                                     ▷ Calculate effective-use-frequency by Eq. (7)

7:      $\mathcal{I} = \text{RandomChoice}(|\mathcal{M}|, B, \text{softmax}(e^{-\hat{c}_i/T}))$      ▷ Sample batch indices from memory

8:      $r = \frac{B}{k|\mathcal{M}|}$                                         ▷ Calculate decay rate

9:      **Update** $c_i \leftarrow (1-r)c_i \quad \forall\, (x_i, y_i) \in \mathcal{M}$      ▷ Decay the sample frequencies

10:     **Update** $C_y \leftarrow (1-r)C_y \quad \forall\, y \in \mathcal{Y}$         ▷ Decay the class frequencies

11:     **Update** $c_i \leftarrow c_i + 1 \quad \forall\, i \in \mathcal{I}$        ▷ Increase sample frequency for selected samples

12:     **Update** $C_{y_i} \leftarrow C_{y_i} + 1 \quad \forall\, i \in \mathcal{I}$       ▷ Increase class frequency for selected samples

13:     $z_t = \{(x_i, y_i) \quad \forall\, i \in \mathcal{I}\}$                        ▷ Obtain training batch $z_t$

14:     $\mathcal{L}(z_t) = \sum_{(x,y) \in z_t} \text{CrossEntropy}(f_\theta(x), y)$         ▷ Calculate loss

15:     $g_{x'_L}(z_t) = \nabla_{x'_L} \mathcal{L}(z_t)$              ▷ Obtain gradient for last feature $x'_L$

16:     **if** $t\%4 = 0$ **then**

17:         **Update** $(trF)_l \leftarrow (1-\alpha)(trF)_l + \alpha \sum (\nabla_{\theta_l} \mathcal{L}(z_t))^2 \quad \forall\, l \in 1, \ldots L$
                                   ▷ Update Fisher every 4 batches

18:         $n^* = 0$                              ▷ No freezing When Fisher update

19:     **else**

20:         $(I/C)_n = \frac{\sum_{l=n+1}^{L}(trF)_l}{(\text{FF}) + \sum_{l=n+1}^{L}(\text{BF})_l} \quad \forall\, n \in 1, \ldots, L$      ▷ Compute $(I/C)$ by Eq. (4)

21:         $\text{BFC}(z_t)_n = \sum_{l=1}^{n}(\text{BF})_l \cdot \max_m (I/C)_m - \frac{|g_{x'_L}(z_t)|^2}{|\bar{g}_{x'_L}|^2} \cdot \sum_{l=1}^{n}(trF)_l \quad \forall\, n \in 1, \ldots, L$
                                   ▷ Compute (BFC) by Eq. (5)

22:         $n^* = \text{argmax}_n \text{BFC}(z_t)_n$        ▷ Determine optimal freezing

23:     **end if**

24:     **Update** $|\bar{g}_{x'_L}| \leftarrow (1-\alpha) \cdot |\bar{g}_{x'_L}| + \alpha \cdot |g_{x'_L}(z_t)|$ ▷ Update expected gradient norm for last feature

25:     $\theta_{S,n^*} = \theta_S \cap \theta_{(n^*+1,\ldots,L)}$         ▷ Use only unfrozen parameters for updating similarity

26:     **Update** $S_{y_i y_j} \leftarrow (1-\alpha)S_{y_i y_j} + \alpha \cdot \text{CosineSimilarity}\left(\nabla_{\theta_{S,n^*}}^{(i)} \mathcal{L}(z_t), \nabla_{\theta_{S,n^*}}^{(j)} \mathcal{L}(z_t)\right)$
     $\forall\, (x_i, y_i), (x_j, y_j) \in z_t, i \neq j$         ▷ Update class similarity using sample-wise gradients

27:     **Update** $\theta_{(n^*+1,\ldots,L)} \leftarrow \theta_{(n^*+1,\ldots,L)} - \mu \cdot \nabla_{\theta_{(n^*+1,\ldots,L)}} \mathcal{L}(z_t)$
                                   ▷ Update the model except frozen layers

28: **end for**

29: **Output** $f_\theta$

---

All layer freezing strategies contribute to reducing computational costs. However, adaptive layer freezing has the least performance decrease. Note that the goal of layer freezing is not to freeze as much as possible but rather to save computational costs while preserving performance.

## A.7   ABLATION STUDY

In addition to the ablation study in the CIFAR-10/100 Gaussian task setup in Sec. 4.3, we ablate the model to investigate the benefit of each of the proposed components in CIFAR-10/100 Disjoint task setup and summarize the results in Tab. 6.

Summing up the effect of the two components, our method outperforms the baseline while using fewer FLOPs than the baseline, each by a noticeable margin.

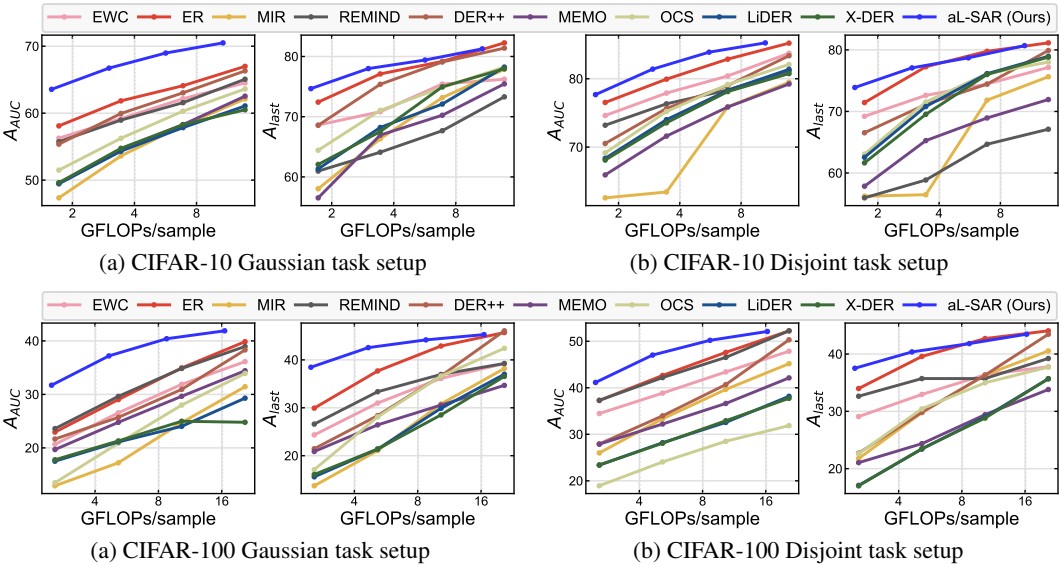

Figure 6: Accuracy on the Gaussian and the Disjoint CL setup in CIFAR-10 and CIFAR-100 for various FLOPs per sample. aL-SAR outperforms all the compared CL methods. The memory budget is fixed to 7.6MB for both CIFAR-10 and CIFAR-100.

| Methods | Gaussian | | | Disjoint | | |
|---|---|---|---|---|---|---|
| | $A_{\text{AUC}}$ ↑ | $A_{\text{last}}$ ↑ | TFLOPs ↓ | $A_{\text{AUC}}$ ↑ | $A_{\text{last}}$ ↑ | TFLOPs ↓ |
| No Freezing | **64.60**±**0.83** | 72.43±0.38 | 171.94 | 79.10±0.44 | **71.77**±**0.57** | 171.94 |
| Random Freezing ($n_{\max} = 16$) | 63.14±0.51 | 70.47±1.15 | 150.56 | 77.69±0.51 | 69.30±1.77 | 150.62 |
| Random Freezing ($n_{\max} = 32$) | 61.79±0.54 | 69.84±0.54 | 122.99 | 77.31±0.17 | 68.89±0.64 | 120.92 |
| Constant Freezing ($n = 8$) | 60.91±0.80 | 67.70±0.83 | 147.12 | 74.99±0.24 | 65.89±0.50 | 147.12 |
| Constant Freezing ($n = 16$) | 53.59±0.60 | 57.31±0.90 | **109.48** | 67.64±0.61 | 55.59±0.29 | **109.48** |
| Linear Freezing ($n_{\max} = 16$) | 63.11±0.90 | 70.00±1.04 | 150.64 | 77.53±0.47 | 68.30±1.52 | 150.64 |
| Linear Freezing ($n_{\max} = 32$) | 62.06±0.90 | 66.95±2.31 | 120.83 | 75.69±0.77 | 64.49±1.22 | 120.83 |
| Adaptive Freezing (Ours) | 64.38±0.32 | **72.57**±**0.79** | 146.80 | **79.75**±**0.38** | 70.70±0.88 | 143.51 |

Table 5: Comparison between adaptive layer freezing and naive freezing in CIFAR-10. The memory budget is 7.6MB.

| Methods | CIFAR-10 | | | CIFAR-100 | | |
|---|---|---|---|---|---|---|
| | $A_{\text{AUC}}$ ↑ | $A_{\text{last}}$ ↑ | TFLOPs ↓ | $A_{\text{AUC}}$ ↑ | $A_{\text{last}}$ ↑ | TFLOPs ↓ |
| Vanilla | 77.10±0.58 | 70.26± 0.91 | 163.73 | 43.59±0.86 | 38.64±0.35 | 245.85 |
| + Freezing | 76.98±0.15 | 70.58±0.63 | **141.50** | 43.20±0.95 | 38.44±0.27 | **225.92** |
| + SAR | 79.10±0.44 | **71.77**±**0.57** | 171.94 | **45.61**±**1.06** | **39.68**±**0.66** | 257.91 |
| + SAR & Freezing (aL-SAR) | **79.75**±**0.38** | 70.70±0.88 | 143.51 | 45.00±1.28 | 39.39±0.62 | 228.14 |

Table 6: Benefits of the proposed components of our method in CIFAR-10 and CIFAR-100 for disjoint task setup. SAR refers to our proposed Similarity-Aware Retrieval method. The memory budget is 7.6MB for CIFAR-10 and 13.44MB for CIFAR-100. CIFAR-10 We train for 1 iter per sample for CIFAR-10 and 1.5 iter per sample for CIFAR-100.

## A.8 ABLATION STUDY OF SIMILARITY-AWARE-RETRIEVAL (SAR)

We report the ablation results on the sub-components of Similarity-Aware Retrieval (SAR) in the Tab. 7 and Tab. 8. Applying only the use-frequency leads to lower performance than random retrieval, since the samples from old tasks which accumulated high use frequencies in the past will be rarely used, causing severe forgetting on past tasks. Applying discounted use-frequency performs marginally better than random retrieval. Considering class similarity with effective use-frequency (*i.e.*, Similarity-Aware Retrieval) further improves performance and outperforms random retrieval. For the ablation study, memory budget is set to 7.6MB for CIFAR-10 and 13.44MB for CIFAR-100. CIFAR-10 We train for 1 iter per sample for CIFAR-10 and 1.5 iter per sample for CIFAR-100.

| Methods | CIFAR-10 | | CIFAR-100 | |
|---|---|---|---|---|
| | $A_{\text{AUC}}$ ↑ | $A_{\text{last}}$ ↑ | $A_{\text{AUC}}$ ↑ | $A_{\text{last}}$ ↑ |
| Vanilla | 60.76±0.11 | 70.08±0.97 | 31.97±0.89 | 37.80±1.30 |
| (+) use-frequency | 56.15±0.43 | 69.40±0.38 | 30.74±1.09 | 36.51±1.32 |
| (+) discounted-use-frequency | 62.72±0.18 | 71.10±0.53 | 33.76±0.67 | 39.29±0.98 |
| (+) effective-use-frequency (SAR) | 64.60±0.83 | 72.43±0.38 | 37.60±0.40 | 42.69±0.18 |

Table 7: **Ablation study of Similarity-Aware Retrieval on Continuous setup.** 'Vanilla' is a simple replay-based method that trains on randomly retrieved batches from a balanced reservoir memory.

| Methods | CIFAR-10 | | CIFAR-100 | |
|---|---|---|---|---|
| | $A_{\text{AUC}}$ ↑ | $A_{\text{last}}$ ↑ | $A_{\text{AUC}}$ ↑ | $A_{\text{last}}$ ↑ |
| Vanilla | 77.10±0.58 | 70.29±0.91 | 43.59±0.86 | 38.64±0.35 |
| (+) use-frequency | 75.40±0.36 | 69.77±1.16 | 42.11±0.81 | 37.83±0.45 |
| (+) discounted-use-frequency | 77.59±0.50 | 71.31±1.44 | 44.45±0.85 | 39.38±0.72 |
| (+) effective-use-frequency (SAR) | 79.10±0.44 | 71.77±0.57 | 45.61±1.06 | 39.68±0.66 |

Table 8: **Ablation study of Similarity-Aware Retrieval on Disjoint setup.** 'Vanilla' is a simple replay-based method that trains on randomly retrieved batches from a balanced reservoir memory.

## A.9    EFFECT OF LAYER FREEZING IN aL-SAR

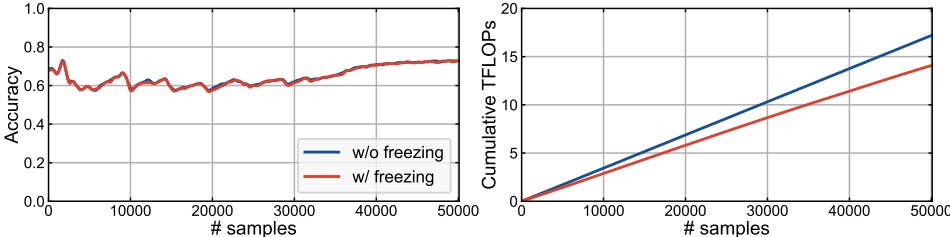

Figure 7: The effect of using the adaptive freezing in aL-SAR, when training 1 iteration per sample in CIFAR-10 Gaussian task setup. The black dotted line is the linear trends from the first 10,000 samples

In Fig. 7, we present the amount of FLOPs adaptive freezing method saves, under the same number of iterations. Our results demonstrate that adaptive freezing can save a substantial amount of training FLOPs up to 22% while maintaining accuracy. Also, considering the FLOPs trends during the first 10,000 samples, represented as the black dotted line, we observe that the ratio of saved FLOPs increases with training progress. This shows that our freezing scheme can capture the training progress, enabling more layers to be frozen once sufficient information has been learned. These observations align with the trends identified in Sec. 4.3 that the ratio of saved FLOPs increases as the total computational budget increases.

## A.10    DETAILED DISTRIBUTION OF BFC ACROSS LAYERS

We plot the BFC values for selected layers and how they evolve during training in Fig. 8. Note that the figure is smoothed to show overall trends of BFC, and the actual values fluctuate depending on the input batch. We observe that (1) BFC values tend to stay negative on average. It indicates that freezing is not beneficial for most batches, because CL imposes a small number of training iterations and continuous streaming of new samples. However, for input batches with little information, BFC temporarily becomes positive to allow freezing. (2) At the task boundaries, we observe a sharp drop of BFC values, which means that the layers are less likely to be frozen so that they can learn from the new, previously unseen data. This shows that BFC correctly handles the shift in informational contents, even though our method does not use any task boundary information. (3) BFC of earlier layers increases as training progresses, and no longer shows significant drop of BFC values at the task boundaries. We believe this is because the earlier layers learn low-level features that are shared across different tasks.

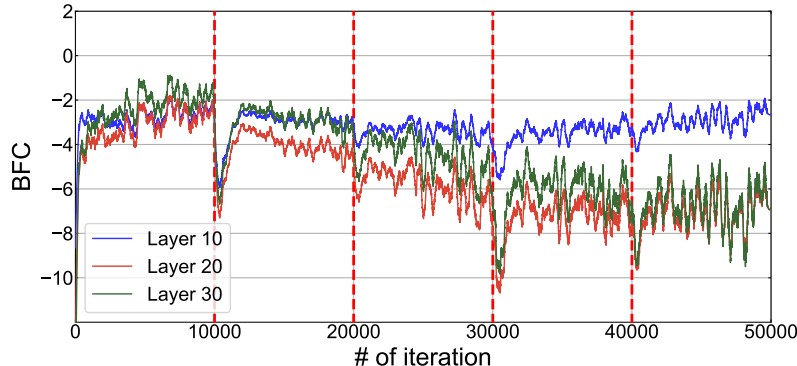

Figure 8: BFC of layer 10, 20, and 30 in ResNet-32 on CIFAR-10 disjoint setup. The red dotted lines indicate the task boundaries.

## A.11 EXPERIMENTAL RESULTS ON TASKS FROM VARIOUS DATASETS

We compare aL-SAR with baselines on 5-datasets (Wang et al., 2022b), which consist of five distinct datasets, to demonstrate its generalizability to the task of continual learning. We report the results in Fig. 9. In the figure, we can see that the x-coordinates of the aL-SAR points are shifted to the left compared to the x-coordinates of the points from other baselines, indicating that less computation was used due to freezing. Note that as the training is done with a higher computational budget, the amount of information that can be learned from the training data relative to the computation decreases. Therefore, the freezing rate increases. In the rightmost point, it uses approximately 20% less computational cost while achieving better performance than other baselines.

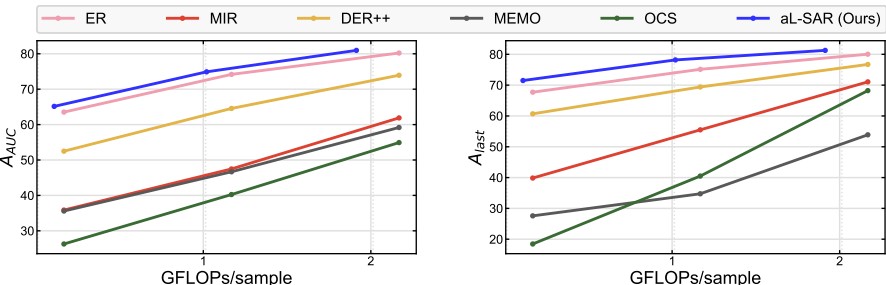

Figure 9: Comparison of baselines in 5-datasets.

## A.12 COMPARISON WITH SOTA EFFICIENT CL METHODS

We compare aL-SAR with with existing SOTA efficient CL methods and summarize the results in Fig. 10. TriRE incorporates distillation through an Exponential Moving Average (EMA) model, necessitating extra forward computation and memory allocation for storage. Consequently, within a total-constrained setup, TriRE retains fewer samples in episodic memory. As a result, TriRE exhibits lower overall performance compared to aL-SAR and SparCL. Especially, TriRE has a significantly lower $A_{\text{AUC}}$ than $A_{\text{last}}$. This is because $A_{\text{AUC}}$ not only includes inference performance at the end of a task but also incorporates anytime inference performance. Specifically, in TriRE, the inference performance at points where all three stages of TriRE, namely Retain, Revise, and Rewind, are not completed tends to be low. Note that both TriRE and SparCL rely on task boundary information, while aL-SAR does not depend on task boundary information, leading to practical utilization in real-world scenarios.

## A.13 DETAILS ON TRAINING MULTI-MODAL LARGE LANGUAGE MODELS

**Datasets.** Beyond the class-incremental learning (CIL) setup, we extend aL to the multi-modal concept-incremental learning (MCIL) setup. Specifically, we evaluate the Bongard-HOI (Jiang et al.,

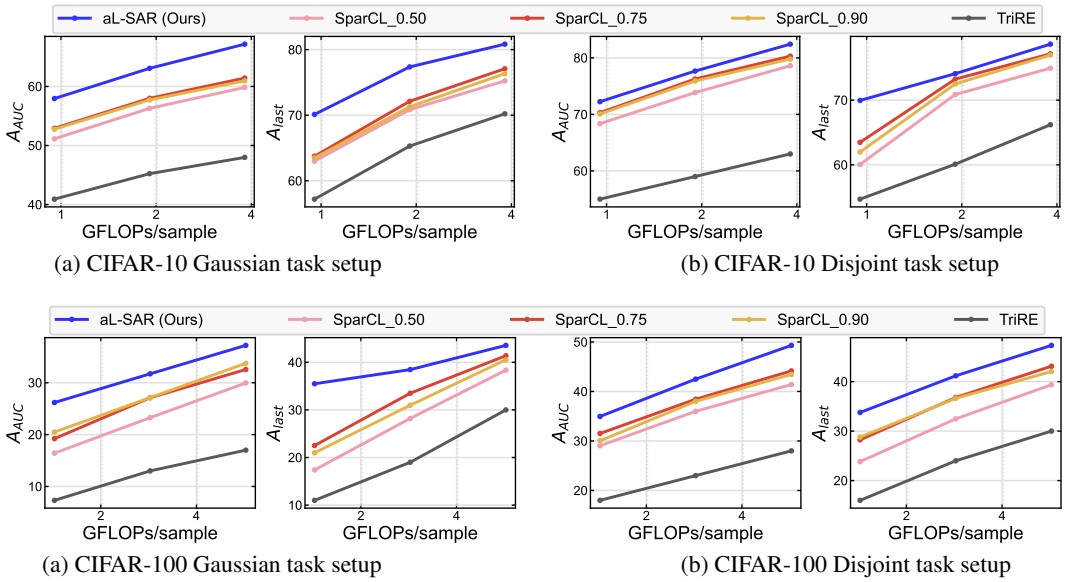

Figure 10: Comparison with efficient CL methods under both memory and computation constrained setup. In SparCL, the underscore represents sparsity. SparCL exhibits its best performance when the sparsity is set to 0.75.

2022) and Bongard-OpenWorld (Wu et al., 2024) benchmarks, which feature multiple 'concepts' (e.g., *ride a bike*, *The top of a snow-covered mountain*) in each benchmark. We split the concepts into 5 disjoint tasks for the MCIL setup. These benchmarks leverage two key attributes of traditional Bongard problems (Depeweg et al., 2018): (1) the capacity for few-shot concept learning and (2) reasoning that is dependent on context. The former entails the ability to derive visual concepts from a limited number of examples, while the latter suggests that the classification of a query image can differ based on the context provided (*i.e.*, the positive and negative support sets). In the Bongard problem, with a positive support set and a negative support set, we tackle two specific tasks: (1) *Which concept is exlusively represented by the positive support set?* and (2) *For a given query image, does it belong to the positive or negative support set?* We denote these tasks as CA (Concept Answering) and P/N, respectively.

**Metrics.** For the Positive/Negative (P/N) task, where answers are either Positive or Negative, we use accuracy as the evaluation metric. Similarly, for the Bongard-HOI CA task, which has simple text answers (e.g., kick a ball), we also use accuracy as the metric. In contrast, for the Bongard-OpenWorld-CA task, where answers are full sentences, we compare the generated sentences from the MLLM model with ground truth sentences. Specifically, following Liu et al. (2023); Yu et al. (2024), we use GPT-3.5 to assess response quality, evaluating helpfulness, relevance, accuracy, and detail. It provides an overall score from 1 to 10, with higher scores indicating better performance. We provide details on the prompts used for GPT-4 evaluation in Sec. A.14. Similar to class-incremental tasks, we also measure the area under the curve (AUC) performance and the final performance within this multi-modal framework.

**Implementation Details.** For the MCIL setup, we assume infinite episodic memory size and use the Adam optimizer with a learning rate of $5 \times 10^{-5}$ and a Constant LR scheduler. The number of images in the support set is set to 2 for each positive and negative support set in Bongard-HOI and Bongard-OpenWorld. We use the LLaVA-1.5-7B model and train it on NVIDIA A100 80GB GPUs.

| Methods | Bongard-HOI-CA | | | Bongard-OpenWorld-CA | | |
|---|---|---|---|---|---|---|
| | $A_{\text{AUC}}$ ↑ | $A_{\text{last}}$ ↑ | TFLOPs ↓ | $A_{\text{AUC}}$ ↑ | $A_{\text{last}}$ ↑ | TFLOPs ↓ |
| No freezing | 74.46±2.77 | **72.41±2.02** | 788.35 | **73.72±0.49** | 73.26±0.56 | 1478.78 |
| aL (Ours) | **75.27±2.15** | 72.31±3.26 | **715.43** (-9.25%) | 73.06±1.15 | **73.92±2.51** | **1401.28** (-5.24%) |

Table 9: **The effect of adaptive freezing on MLLM training.** We use LLaVA-1.5-7B model with LoRA.

In Tab. 3 in Sec. 4.2, we empirically demonstrate the effectiveness of aL in the Bongard-HOI-P/N and Bongard-OpenWorld-P/N tasks.

**Effect of aL in CA tasks.** In addition to the P/N tasks presented in Table 3, we also apply aL to the CA tasks and summarize the results in Table 9. Similar to the P/N tasks, aL effectively reduces training costs in CA tasks. However, compared to the relatively simple P/N tasks, which involve binary classification, the CA tasks require generating text or sentence-level answers, making them more complex. As a result, aL leads to a relatively smaller reduction in FLOPs in CA tasks, compared to P/N tasks. It shows that aL adaptively freezes fewer layers for more complex tasks that require more training.

Notably, our proposed adaptive layer freezing method operates independently of downstream tasks, model architecture, and task identity information, thus we expect it to be applicable across a wide range of tasks and model architectures.

## A.14 DETAILS OF PROMPTS USED FOR GPT-3.5 EVALUATION

We use GPT-3.5 to assess the quality of predicted sentences against ground truth sentences using the following prompts:

> [Ground truth]: 'Ground-Truth sentence'
> [Prediction]: 'Prediction sentence'
> Compare the ground truth and prediction from AI models, to give a correctness score for the prediction. The user asks the question about observing an image or multiple images. The images replace <image> tokens in [Input]. Please rate the relevance and accuracy of [Assistant] compared to [Ground-Truth]. The assistant receives an overall score on a scale of 1 (totally wrong) to 10 (totally right), where a higher score indicates better overall performance. Please first provide a comprehensive explanation of your evaluation explanation. In the subsequent line, please provide a single line containing the score for Assistant, in the format of Score: <an integer value on a scale 1 to 10>

## A.15 COMPARISON BETWEEN aL AND OTHER FREEZING METHODS

In addition to the quantitative comparison of aL with freezing methods in the continuous setup, we also compare them in the disjoint setup. The results are summarized in Tab. 10. Similar to the continuous setup, aL effectively reduces the training FLOPs in disjoint setup while maintaining performance. In contrast, REMIND (Hayes et al., 2020), PTLF (Yang et al., 2023), and EGERIA (Wang et al., 2023) result in negligible reductions in training costs, indicating less freezing. The reasons are as follows: (1) EGERIA freezes layers where the difference between the outputs of the current model and the reference model has been marginal over recent $n$ iterations. The reference model is periodically updated using the training model to evaluate layer plasticity. This approach can be effective in joint training scenarios, as the training loss typically converges to zero over time, causing the layers to stabilize. However, in online CL, training a current model with new data can cause the model to diverge from the reference model, resulting in less effective layer freezing. (2) PTLF freezes layers with top-$k$ task correlation ratio. However, since it freezes intermediate layers rather than layers 1 to $n$, it cannot fully reduce the backward FLOPs of the frozen layers as the gradients for layer input should still be calculated for backpropagation.

| Methods | CIFAR-10 | | | CIFAR-100 | | |
|---|---|---|---|---|---|---|
| | $A_{AUC}$ ↑ | $A_{last}$ ↑ | TFLOPs ↓ | $A_{AUC}$ ↑ | $A_{last}$ ↑ | TFLOPs ↓ |
| No freezing | $79.10 \pm 0.44$ | $\mathbf{71.77 \pm 0.57}$ | 171.94 | $\mathbf{53.38 \pm 1.13}$ | $\mathbf{49.43 \pm 0.72}$ | 515.82 |
| REMIND | $73.20 \pm 0.67$ | $62.95 \pm 0.87$ | 151.31 (-12.0%) | $48.14 \pm 1.30$ | $41.28 \pm 0.14$ | 453.92 (-12.0%) |
| PTLF | $79.29 \pm 0.36$ | $70.50 \pm 0.43$ | 164.38 (-4.4%) | $53.22 \pm 1.07$ | $49.25 \pm 0.65$ | 497.77 (-3.5%) |
| EGERIA | $78.70 \pm 0.54$ | $70.72 \pm 0.68$ | 169.52 (-1.4%) | $52.95 \pm 0.77$ | $48.88 \pm 0.46$ | 502.41 (-2.6%) |
| aL (Ours) | $\mathbf{79.75 \pm 0.38}$ | $70.70 \pm 0.88$ | $\mathbf{143.51}$ (-16.5%) | $53.33 \pm 1.21$ | $48.91 \pm 0.87$ | $\mathbf{425.65}$ (-17.5%) |

Table 10: **Comparison between our proposed adaptive layer freezing and other freezing methods.** We compare them in both the CIFAR-10 disjoint setup and the CIFAR-100 disjoint setup. We train for 1 iter per sample for CIFAR-10 and 3 iter per sample for CIFAR-100.

### A.16 Applying Adaptive Layer Freezing to Attention-based Model

We investigate the effect of the proposed adaptive layer freezing not only in ResNet but also in attention-based models such as the Vision Transformer(ViT) (Dosovitskiy et al., 2020).We compare the freezing effects for both ViT-base and ViT-large models, considering those pretrained on ImageNet-1K and those trained from scratch. The results are summarized in Tab.11 and Tab. 12, respectively.

When using a pretrained model, the adaptive layer freezing reduces the computational cost by nearly 15% with minimal impact on $A_{\text{AUC}}$ and $A_{last}$, compared to the Vanilla training without freezing. Since pretrained models have already been sufficiently trained on a large dataset, the amount of information that the model will learn from the training data may be relatively small compared to training from scratch. Thus, it leads to the freezing of many layers by adaptive layer freezing. This not only reduces computational costs but also ensures high performance, since the model is updated only in truly informative batches, thus preserving the advantages of pretrained initialization.

In the case of training from scratch, the decrease in TFLOPs is significantly small compared to using a pretrained model, which implies that the layers did not freeze much. This is due to the large model capacity of ViT and the small number of training iterations in online CL, which leads to a severe underfitting of the model when training from scratch. Specifically, Tab. 11 shows that when training from scratch, the accuracy only reaches around 30% even at the end of the training ($A_{last}$). Thus, since the model is not sufficiently trained yet, the adaptive layer freezing scheme tends to freeze fewer layers so that the model can learn more information. This shows that the proposed adaptive freezing method can indeed provide a reasonable freezing strategy.

| Methods | CIFAR10 | | | | | | CIFAR100 | | | | | |
| | Pretrained | | | From Scratch | | | Pretrained | | | From Scratch | | |
| | $A_{\text{AUC}}$ ↑ | $A_{\text{last}}$ ↑ | TFLOPs ↓ | $A_{\text{AUC}}$ ↑ | $A_{\text{last}}$ ↑ | TFLOPs ↓ | $A_{\text{AUC}}$ ↑ | $A_{\text{last}}$ ↑ | TFLOPs ↓ | $A_{\text{AUC}}$ ↑ | $A_{\text{last}}$ ↑ | TFLOPs ↓ |
|---|---|---|---|---|---|---|---|---|---|---|---|---|
| Vanilla | 57.85±1.16 | 61.43±0.68 | 4,044.25 | 33.13±3.15 | 28.58±4.64 | 4,044.25 | 40.43±0.31 | 45.03±0.06 | 6066.4 | 14.95±5.83 | 15.61±7.58 | 6066.4 |
| + Adaptive Freezing | 58.70±1.58 | 60.73±1.52 | 3,466.42 | 33.34±3.44 | 30.44±6.22 | 3,926.65 | 41.25±2.95 | 46.58±1.71 | 5224.05 | 21.13±0.74 | 24.78±0.21 | 5828.75 |

Table 11: Effect of layer freezing in ViT-base. We used CIFAR-10 and CIFAR-100 as the dataset, Gaussian Setup for setup. The memory budget is 334.76MB for both CIFAR-10 and CIFAR-100.

| Methods | Disjoint | | | | | | Gaussian | | | | | |
| | Pretrained | | | From Scratch | | | Pretrained | | | From Scratch | | |
| | $A_{\text{AUC}}$ ↑ | $A_{\text{last}}$ ↑ | TFLOPs ↓ | $A_{\text{AUC}}$ ↑ | $A_{\text{last}}$ ↑ | TFLOPs ↓ | $A_{\text{AUC}}$ ↑ | $A_{\text{last}}$ ↑ | TFLOPs ↓ | $A_{\text{AUC}}$ ↑ | $A_{\text{last}}$ ↑ | TFLOPs ↓ |
|---|---|---|---|---|---|---|---|---|---|---|---|---|
| Vanilla | 69.96±3.58 | 55.03±4.46 | 14,316.37 | 41.27±0.79 | 19.52±1.47 | 14,316.37 | 58.82±2.17 | 60.11±4.81 | 14,3163.73 | 28.05±1.74 | 24.17±2.69 | 14,316.37 |
| + Adaptive Freezing | 70.78±4.70 | 58.34±4.06 | 11,002.60 | 41.63±1.09 | 21.19±1.90 | 13,802.64 | 64.15±2.70 | 73.12±0.18 | 11,306.60 | 28.93±2.08 | 24.34±1.72 | 14,017.93 |

Table 12: Effect of layer freezing in ViT-Large. We used CIFAR-10 and CIFAR-100 as the dataset, Gaussian Setup for setup. The memory budget is 1.20GB for both CIFAR-10 and CIFAR-100.

### A.17 Applying aL-SAR across Diverse Network Architectures

aL-SAR can be applied to any feedforward neural network, as long as layers can be defined, including CNNs and Vision Transformers (Dosovitskiy et al., 2020), as shown in Sec. A.16. Note that since our layer freezing methods require evaluating the information gained and FLOPs used by individual layers, a network should be dissected into layers.

| Model | CIFAR-10 | | | | | | CIFAR-100 | | | | | |
| | Disjoint | | | Gaussian | | | Disjoint | | | Gaussian | | |
| | $A_{\text{AUC}}$ ↑ | $A_{\text{last}}$ ↑ | TFLOPs ↓ | $A_{\text{AUC}}$ ↑ | $A_{\text{last}}$ ↑ | TFLOPs ↓ | $A_{\text{AUC}}$ ↑ | $A_{\text{last}}$ ↑ | TFLOPs ↓ | $A_{\text{AUC}}$ ↑ | $A_{\text{last}}$ ↑ | TFLOPs ↓ |
|---|---|---|---|---|---|---|---|---|---|---|---|---|
| Baseline | 78.11±0.18 | 73.44±0.91 | 506.54 | 58.95±0.11 | 74.59±0.27 | 506.54 | 45.37±1.14 | 35.68±0.35 | 759.81 | 32.91±0.64 | 34.86±0.68 | 759.81 |
| aL-SAR (Ours) | 81.25±0.23 | 75.28±0.29 | 418.19 | 66.44±0.18 | 77.08±1.09 | 437.34 | 50.15±0.92 | 38.57±0.55 | 647.77 | 41.88±0.58 | 41.65±1.02 | 668.11 |

Table 13: Comparison between Baseline and aL-SAR on Disjoint and Gaussian in CIFAR-10 and CIFAR-100 with naive 8-CNN layers. The baseline refers to removing the two components of aL-SAR: similarity-aware retrieval and adaptive layer freezing.

To analyze the effect of the number of layers in aL-SAR, we first conduct experiments with ResNet-20 and ResNet-56 on CIFAR-10 and CIFAR-100, in addition to ResNet-32 and ResNet-18 as used in the main results. As shown in Fig. 11 and Fig. 12, aL-SAR consistently outperforms baseline, which refers to removing two components of aL-SAR: adaptive layer freezing and similarity-aware retrieval, irrespective of the number of layers. We also compare aL-SAR with other CL baselines

| Model | CIFAR-10 | | | | | | CIFAR-100 | | | | | |
|---|---|---|---|---|---|---|---|---|---|---|---|---|
| | Disjoint | | | Gaussian | | | Disjoint | | | Gaussian | | |
| | $A_{\text{AUC}}$ ↑ | $A_{\text{last}}$ ↑ | TFLOPs ↓ | $A_{\text{AUC}}$ ↑ | $A_{\text{last}}$ ↑ | TFLOPs ↓ | $A_{\text{AUC}}$ ↑ | $A_{\text{last}}$ ↑ | TFLOPs ↓ | $A_{\text{AUC}}$ ↑ | $A_{\text{last}}$ ↑ | TFLOPs ↓ |
| Baseline | 75.45±0.86 | 72.09±0.84 | 1236.51 | 54.59±0.62 | 71.43±0.46 | 1236.51 | 40.86±1.62 | 33.56±0.98 | 1854.78 | 28.49±0.38 | 32.01±0.16 | 1854.78 |
| aL-SAR (Ours) | **79.43±0.34** | **73.62±1.31** | **1025.14** | **62.71±0.73** | **74.62±2.06** | **1082.22** | **46.22±1.38** | **39.70±0.13** | **1631.72** | **38.36±1.04** | **41.35±1.51** | **1679.66** |

Table 14: Comparison between Baseline and aL-SAR on Disjoint and Gaussian in CIFAR-10 and CIFAR-100 with naive 16-CNN layers. The baseline refers to removing the two components of aL-SAR: similarity-aware retrieval and adaptive layer freezing.

| Model | CIFAR-10 | | | | | | CIFAR-100 | | | | | |
|---|---|---|---|---|---|---|---|---|---|---|---|---|
| | Disjoint | | | Gaussian | | | Disjoint | | | Gaussian | | |
| | $A_{\text{AUC}}$ ↑ | $A_{\text{last}}$ ↑ | TFLOPs ↓ | $A_{\text{AUC}}$ ↑ | $A_{\text{last}}$ ↑ | TFLOPs ↓ | $A_{\text{AUC}}$ ↑ | $A_{\text{last}}$ ↑ | TFLOPs ↓ | $A_{\text{AUC}}$ ↑ | $A_{\text{last}}$ ↑ | TFLOPs ↓ |
| Baseline | 68.28±0.95 | 65.18±1.66 | 2096.46 | 46.00±1.55 | 60.30±2.39 | 2696.46 | 27.81±0.30 | 23.11±0.78 | 4044.71 | 17.69±1.16 | 21.69±1.43 | 4044.71 |
| aL-SAR (Ours) | **73.42±0.41** | **66.51±0.23** | **2304.73** | **53.21±3.01** | **66.97±3.87** | **2495.75** | **29.80±0.96** | **26.05±1.00** | **3910.97** | **24.68±2.10** | **29.69±2.65** | **3940.56** |

Table 15: Comparison between Baseline and aL-SAR on Disjoint and Gaussian in CIFAR-10 and CIFAR-100 with naive 32-CNN layers. The baseline refers to removing the two components of aL-SAR: similarity-aware retrieval and adaptive layer freezing.

and summarize the results in Fig.13 and Fig. 14, respectively. aL-SAR consistently outperforms other methods in both shallower and deeper networks, showing that aL-SAR is robust in various model sizes.

Moreover, we perform additional experiments on the naive CNN without skip connection, *i.e.*, consisting only of convolution, batch normalization, activation, and fully connected layers. We report the result in Tab. 13, Table Tab. 14, and Tab. 15 for 8-layer, 16-layer, and 32-layer CNNs, respectively. In the table, aL-SAR improves performance and reduces computational cost compared to baseline also when using a simple CNN, regardless of the number of layers. Note that a deeper CNN shows lower performance due to the absence of skip connections, as reported in [1] (ResNet).

However, our proposed adaptive layer freezing cannot apply to recurrent neural networks (Sherstinsky, 2020) since the gradient of a layer affects not only the preceding layers but also the subsequent layers (Rotman & Wolf, 2021), while in the feedforward network, the gradient of a layer influences only the preceding layers.

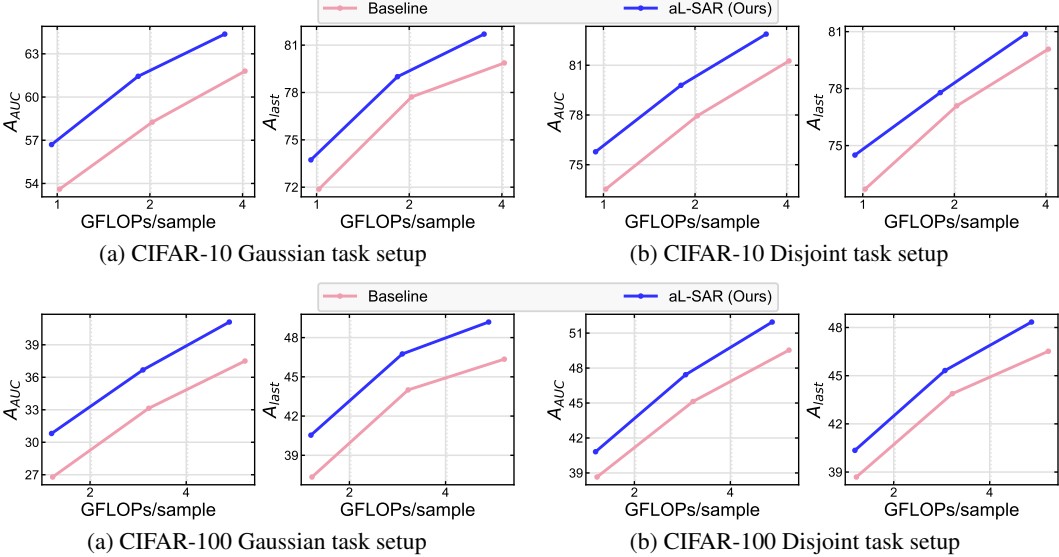

Figure 11: Comparison of the baseline and aL-SAR on Gaussian and Disjoint CL setup in CIFAR-10 and CIFAR-100 with ResNet-20. The baseline refers to removing the two components of aL-SAR, *i.e.*, adaptive layer freezing and similarity-aware retrieval, from aL-SAR itself.

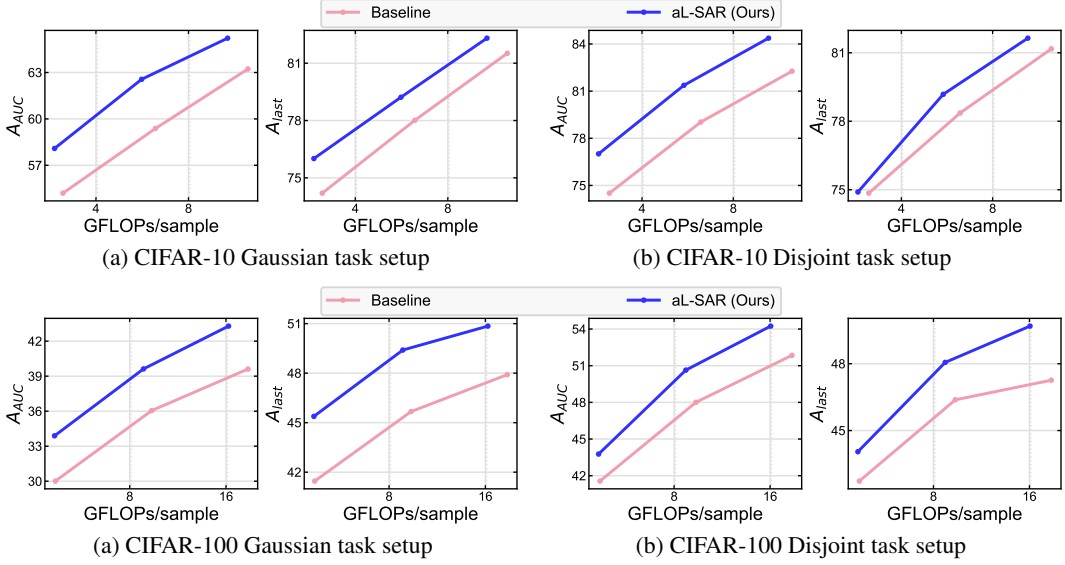

Figure 12: Comparison of the baseline and aL-SAR Gaussian and Disjoint CL setup in CIFAR-10 and CIFAR-100 with ResNet-56. The baseline refers to removing the two components of aL-SAR, *i.e.*, adaptive layer freezing and similarity-aware retrieval, from aL-SAR itself.

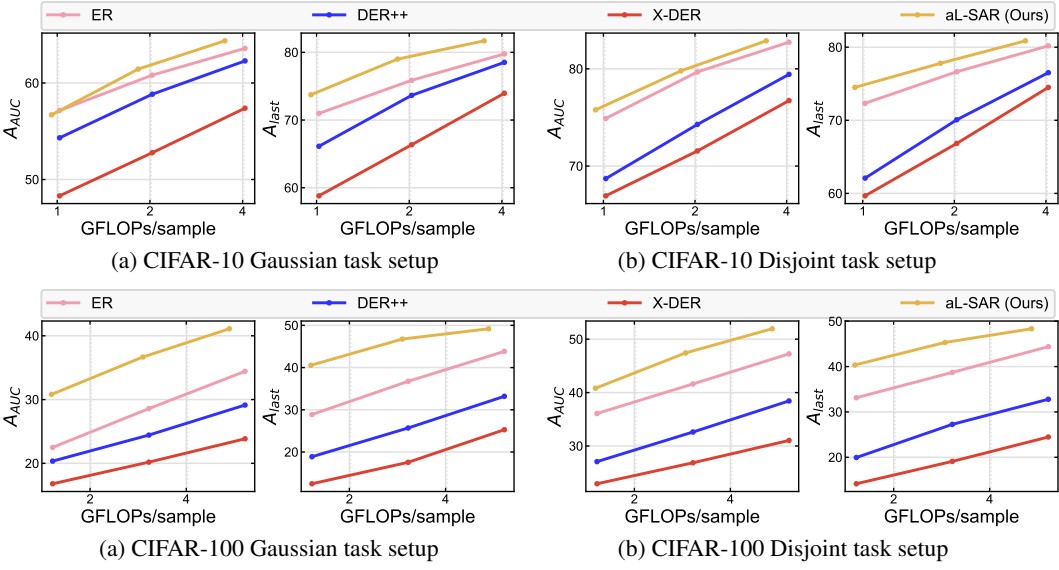

Figure 13: Comparison of CL methods on Gaussian and Disjoint CL setup in CIFAR-10 and CIFAR-100 with ResNet-20.

## A.18    DETAILED INFORMATION ON THE CALCULATION OF $\mathcal{S}_{y_i, y_j}$

We calculate the cosine similarity of the gradients between all pairs of samples in the training batch and update the EMA estimate of class-wise gradients, which are then used for calculating $S_{y_i y_j}$. To further reduce the computational cost of calculating similarity, we use only $0.05\%$ of the model parameters for the calculation of similarity, since the gradient distribution of the subset of randomly selected weights is similar to the gradient of the entire weight set (Li et al., 2022).

Specifically, we randomly select 0.05% of the model parameters across all layers before training. During training, we use the gradients of the pre-selected, which are unfrozen by adaptive layer freezing, to update the class-wise gradients. Specifically, we employ an EMA estimation of the class-

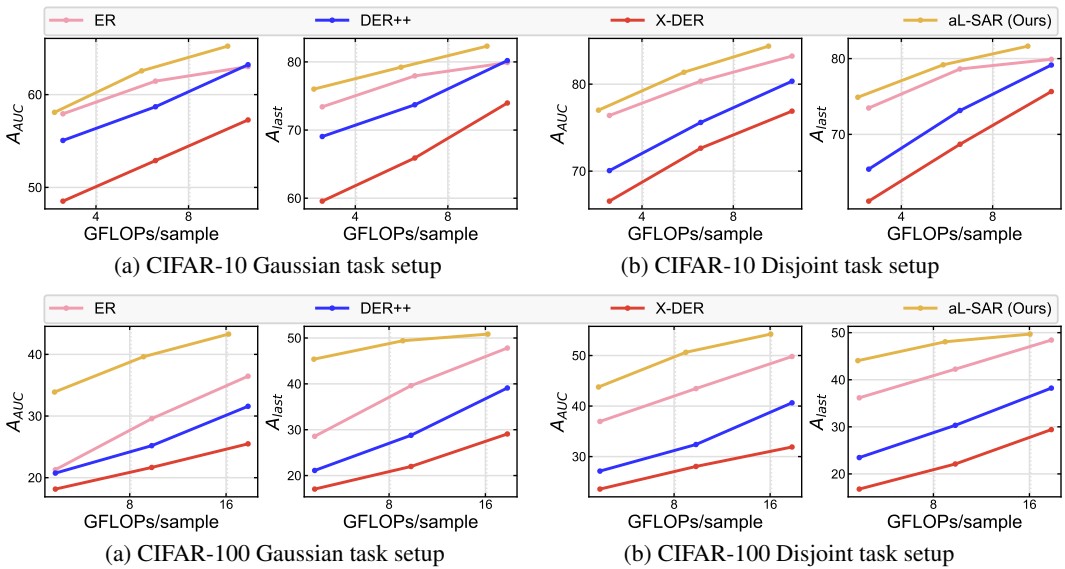

Figure 14: Comparison of CL methods on Gaussian and Disjoint CL setup in CIFAR-10 and CIFAR-100 with ResNet-56.

wise gradients, updating them every batch iteration using the gradients of selected parameters in an EMA manner. While not all EMA-estimated class-wise gradients are updated every iteration (since gradients of frozen layers are not computed), all elements of the estimated class-wise gradients are updated every $m$ iterations, as we unfreeze all layers every $m$ iterations, as mentioned in Sec. A.1. To form the similarity matrix, we first compute the element-wise product of class-wise gradients and then average them over all class-wise pairs.

### A.19 COMPARISON WITH RETRIEVAL-BASED CL METHODS

To train models with informative training batches, recent studies propose retrieving batches from episodic memory, such as MIR (Aljundi et al., 2019a) and ASER (Shim et al., 2021). However, calculating the loss in MIR and the adversarial shapely value in ASER requires additional forward and backward processes, leading to high computational requirements. In contrast, our proposed similarity-aware retrieval methods only utilize the gradient vector, which can be obtained naturally during the training process, incurring no additional cost. Therefore, as depicted in Fig. 15, aL-SAR outperforms MIR and ASER under fixed computational budgets.

### A.20 EFFECT OF TEMPERATURE $T$

Temperature is selected by a hyperparameter search on CIFAR-10. The lower the temperature, the more the retrieval focuses on samples with low effective use-frequency, enabling a faster gain in knowledge. However, since we need diverse samples to maintain an accurate estimate of similarity, a too low temperature would also hinder the performance. Thus, we select an adequate temperature via a hyperparameter search. We report the result of the hyperparameter search in Fig. 16. Note that while a too high or low temperature results in a diminished performance, there is a wide range of temperature values that show stable performances.

### A.21 DETAILS ABOUT MEASURING FLOPS

With the exception of the simplest baseline, ER, the other baseline methods involve additional computational costs. REMIND, MIR, OCS, X-DER, and MEMO require additional forward/backward processes of the model, resulting in a notable increase in the relative FLOPs compared to ER. On the contrary, for DER, EWC, and aL-SAR, additional computational cost is neglible as they do not involve model forward and backward processes *i.e.*, the relative FLOPs is approximately 1. We

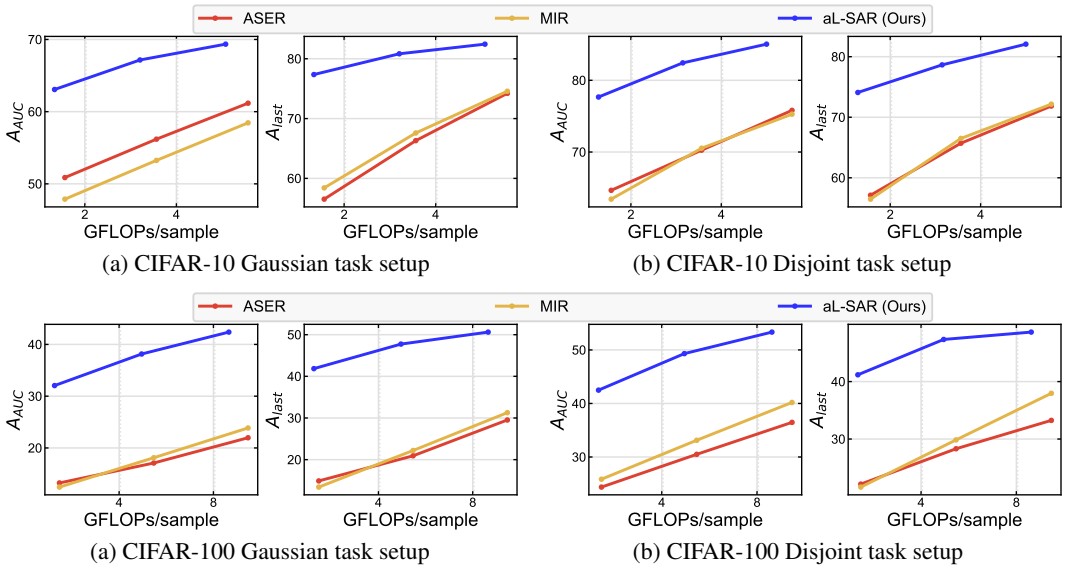

Figure 15: Comparison of various retrieval-based CL methods on Gaussian and Disjoint CL setup in CIFAR-10 and CIFAR-100.

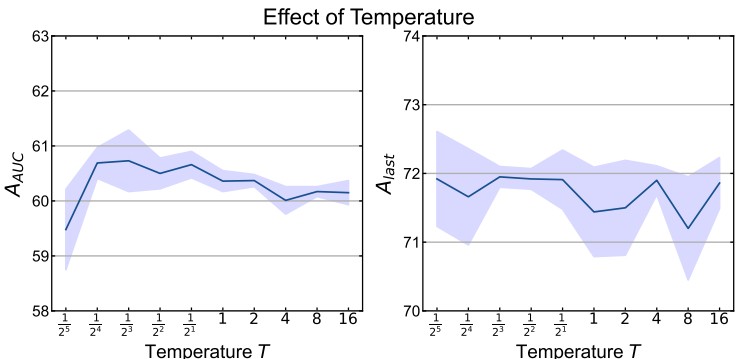

Figure 16: Experiment for the effect of temperature $T$ in the CIFAR-10 Gaussian scheduled setup. We use ResNet-32 as a backbone

compare the types of additional computational costs and relative flops compared to ER (Rolnick et al., 2019) for each method in Tab. 16.

Although aL-SAR requires various types of computation, these amounts to significantly fewer operations compared to the costs incurred during model forward and backward passes. In addition, since aL-SAR freezes layers depending on the input batch, we provided a range for backward FLOPs. Note that Remind incurs additional costs during base initialization to convert all memory data into features and perform product quantization. However, since this process occurs only once during the model training process and not in every iteration, it is not included in the table.

## A.22  Experimental Results in Memory Infinite Setup

aL-SAR demonstrates even stronger performance with an unlimited memory budget setup in various benchmarks such as CIFAR-10, CIFAR-100, and ImageNet-1K, as shown in Fig. 17. We believe it is because the effect of the retrieval strategy is more significant with unlimited memory than with limited memory. Provided that the retrieval strategy successfully distinguishes useful samples, retrieving from the full past data (unlimited memory) is likely to yield more useful samples than retrieving only from a small portion of past data (limited memory). Thus, strong performance in the

| Methods | Type of Computation | FLOPs formulation | FLOPs/sample (TFLOPs) | Total FLOPs/sample (TFLOPs) | Relative FLOPs to ER |
|---|---|---|---|---|---|
| ER | Forward FLOPs | Measured by ptflops | 0.57 | 1.71 | 1 |
|  | Backward FLOPs | Measured by ptflops | 1.14 |  |  |
|  | Model Updates (Adam) | $12 \times$ (# parameters) | $5.52 \times 10^{-6}$ |  |  |
| DER | Forward FLOPs | Measured by ptflops | 0.57 | 1.71 | $\approx 1$ |
|  | Backward FLOPs | Measured by ptflops | 1.14 |  |  |
|  | Model Updates (Adam) | $12 \times$ (# parameters) | $5.52 \times 10^{-6}$ |  |  |
|  | Calculation of Distillation Loss | $2 \times$ (# classes) | $1.0 \times 10^{-11}$ |  |  |
| X-DER | Forward FLOPs | Measured by ptflops | 0.57 | 3.99 | 2.33 |
|  | Backward FLOPs | Measured by ptflops | 1.14 |  |  |
|  | Model Updates (Adam) | $12 \times$ (# parameters) | $5.52 \times 10^{-6}$ |  |  |
|  | Forwarding Inputs with Different Augmentations for Contrastive Learning | $4 \times$ (Forward FLOPs) | 2.28 |  |  |
| OCS | Forward FLOPs | Measured by ptflops | 0.57 | 3.42 | $\approx 2$ |
|  | Backward FLOPs | Measured by ptflops | 1.14 |  |  |
|  | Model Updates (Adam) | $12 \times$ (# parameters) | $5.52 \times 10^{-6}$ |  |  |
|  | Coreset sampling | (Forward FLOPs + Backward FLOPs) | 1.71 |  |  |
| ER-MIR | Forward FLOPs | Measured by ptflops | 0.57 | 7.05 | 4.12 |
|  | Backward FLOPs | Measured by ptflops | 1.14 |  |  |
|  | Model Updates (Adam) | $12 \times$ (# parameters) | $5.52 \times 10^{-6}$ |  |  |
|  | Forward FLOPs for Candidates | $\frac{(\#\text{candidates})}{(\text{batchsize})} \times 0.57$ | 1.78 |  |  |
|  | Backward FLOPs for Candidates | $\frac{(\#\text{candidates})}{(\text{batchsize})} \times 1.14$ | 3.56 |  |  |
| MEMO | Forward FLOPs | Measured by ptflops | 0.57 | 2.55 | 1.49 |
|  | Backward FLOPs | Measured by ptflops | 1.14 |  |  |
|  | Model Updates (Adam) | $12 \times$ (# parameters) | $5.52 \times 10^{-6}$ |  |  |
|  | Forward cost for Expanded Network | (# of tasks) $\times 0.03$ | 0.15 |  |  |
|  | Backward cost for Expanded Network | (# of tasks) $\times 0.07$ | 0.35 |  |  |
| LiDER | Forward FLOPs | Measured by ptflops | 0.57 | 3.99 | 2.33 |
|  | Backward FLOPs | Measured by ptflops | 1.14 |  |  |
|  | Model Updates (Adam) | $12 \times$ (# parameters) | $5.52 \times 10^{-6}$ |  |  |
|  | Forwarding Inputs with Different-Augmentations for Contrastive Learning | $4 \times$ (Forward FLOPs) | 2.28 |  |  |
|  | Calculation of Lipschitz Constant | (batchsize) $\times$ f.shape[0] * f.shape[0] $\times$ (2 - f.shape[1] - 1) | $4.19 \times 10^{-6}$ |  |  |
| CCL-DC | Forward FLOPs | Measured by ptflops | 0.57 | 7.98 | 4.66 |
|  | Backward FLOPs | Measured by ptflops | 1.14 |  |  |
|  | Model Updates (Adam) | $12 \times$ (# parameters) | $5.52 \times 10^{-6}$ |  |  |
|  | Forwarding Inputs with Different-Augmentations for Collaborative Learning | $9 \times$ (Forward FLOPs) + (Backward FLOPs) | 6.27 |  |  |
| CAMA | Forward FLOPs | Measured by ptflops | 0.57 | 1.71 | $\approx 1$ |
|  | Backward FLOPs | Measured by ptflops | 1.14 |  |  |
|  | Model Updates (Adam) | $12 \times$ (# parameters) | $5.52 \times 10^{-6}$ |  |  |
|  | Calculation of Distillation Loss | $2 \times$ (# classes) | $1.0 \times 10^{-11}$ |  |  |
|  | Logit Updating | $3 \times$ (# classes) | $1.5 \times 10^{-11}$ |  |  |
| REMIND | Forward FLOPs (before base initialization) | Measured by ptflops | 0.57 | 1.71 | 1 |
|  | Backward FLOPs (before base initialization) | Measured by ptflops | 1.14 |  |  |
|  | Model Updates (Adam - before base initialization) | $12 \times$ (# parameters) | $5.52 \times 10^{-6}$ |  |  |
|  | Forward FLOPs (after base initialization) | $0.78 \times$ (Forward FLOPs for whole model) | 0.44 | 1.32 | 0.7 |
|  | Backward FLOPs (after base initialization) | $0.78 \times$ (Backward FLOPs for whole model) | 0.88 |  |  |
|  | Model Updates (Adam - after base initialization) | $0.78 \times$ (Model updates for whole model) | $5.52 \times 10^{-6}$ |  |  |
| aL-SAR | Forward FLOPs | Measured by ptflops | 0.57 | $0.57 \sim 1.71$ | $0.33 \sim 1$ |
|  | Backward FLOPs | Measured by ptflops (depends on freezing) | $8.2 \times 10^{-8} \sim 1.14$ |  |  |
|  | Model Updates (Adam) | $12 \times$ (# parameters) | $5.52 \times 10^{-6}$ |  |  |
|  | Calculation of Class-wise similarity | $(5 \times$ (# parameters) $\times 0.0005 + 3)$ $\times$ (batchsize | $5.5 \times 10^{-8}$ |  |  |
|  | Calculation of Fisher Information | $2 \times$ (# parameters) + $3 \times$ (# layers) | $9.2 \times 10^{-7}$ |  |  |
|  | Calculation of BFC | (# layers) $\times 3 + \text{len}(x_L) \times 2$ | $1.1 \times 10^{-9}$ |  |  |
|  | Calculation of Retrieval Probability | $4 \times |M| +$ (# classes)$^2$ + (# classes) | $8.11 \times 10^{-9}$ |  |  |
|  | Frequency Update | (batchsize) $\times 2 +$ (# classes) $+ |M|$ | $2.0 \times 10^{-9}$ |  |  |

Table 16: Details of the additional computational budget. To measure forward/backward FLOPs of the model, we use ptflops[1], which is a widely used Python library to calculate FLOPs. FLOPs from other operations were manually calculated.

unlimited-memory setup indicates that our similarity-aware retrieval effectively distinguishes useful samples.

Interestingly, aL-SAR freezes fewer layers in the unlimited memory setup than the limited memory setup. In the unlimited memory setup, the model can learn more knowledge from the samples stored in episodic memory than in the limited memory setup, so our adaptive freezing scheme chooses to freeze fewer layers so that the model can acquire more knowledge. It shows that our adaptive freezing method works also in the unlimited memory setup without the need to modify.

## A.23 Details about the Memory bBudget in Total-constrained CL

The memory budget is allocated to episodic memory, model parameters, and additional memory costs specific to each CL algorithm, such as classwise similarities and logits. In this section, $\mathcal{B}$ denotes the additional memory budget, $S(|\mathcal{B}|)$ denotes the size of the additional memory budget (in MB), $\mathcal{E}$ denotes episodic memory, and $|\mathcal{E}|$ represents the number of stored instances in $\mathcal{E}$.

In our total-constrained setup, the memory budget is restricted to the cost of storing 7.6MB, 13.44MB and 25.12 MB in CIFAR-10/100. Since storing the ResNet-32 model parameters requires memory cost equivalent to saving 603 instances of CIFAR-100 images (463,504 floats $\times$ 4 bytes/float $\div$ ($3 \times 32 \times 32$) bytes/image $\approx$ 603 instances), for methods that store the model for distillation or

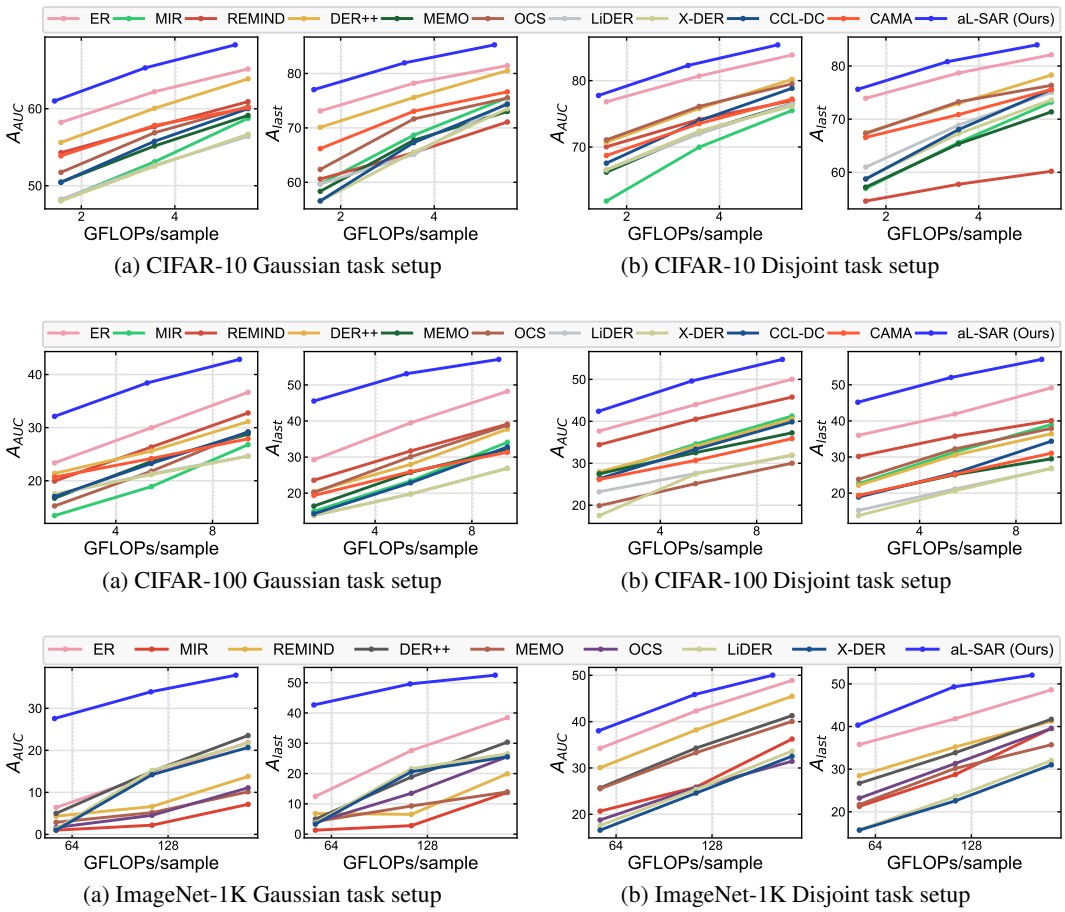

Figure 17: $A_{last}$ and $A_{\mathrm{AUC}}$ on Gaussian and Disjoint CL setup in CIFAR-10 and CIFAR-100 for a wide range of FLOPs per sample with infinite memory budget. We employ ResNet-32 for CIFAR-10/100 and ResNet-18 for ImageNet-1K as a backbone.

regularization, we subtract the memory cost of the model parameters from the episodic memory size (Zhou et al., 2023). In ImageNet and CLEAR-10, we use the ResNet-18 model and apply the same policy of subtracting model parameters and logits from the memory budget as mentioned above.

ER does not require additional memory beyond episodic memory. Similarly, MIR (Aljundi et al., 2019a) and ASER (Shim et al., 2021) do not require additional memory despite being computationally heavy.

On the contrary, EWC (Kirkpatrick et al., 2017) requires storing the previous model parameters and the parameter-wise Fisher Information(FI) for all parameters. Therefore, we subtract the memory cost of storing two models from the episodic memory size. Similarly, BiC (Wu et al., 2019) also stores the previous model for distillation, $|\mathcal{E}|$ was reduced as much as the size of the model. For example, with a total memory budget of 7.6MB and a ResNet-32 model type in CIFAR-100, ER can store up to 2000 instances in $\mathcal{E}$, while EWC is limited to storing only 794 (= 2000 - 2×603) instances. Similarly, BiC can store only 1397 (= 2000 - 603) instances in $\mathcal{E}$.

Some methods incur additional memory costs other than episodic memory or model parameters. We handle such costs in a similar way by reducing the episodic memory size by the number of samples equivalent to the additional memory cost. For example, DER (Buzzega et al., 2020) uses the previous logits of the samples for distillation, so we subtract the cost of storing the logits from the episodic memory size. More specifically, DER needs additional storage which size is $|\mathcal{E}| \times d_l \times$ 4 bytes/float, where $d_l$ denotes logit dimension, which is 100 in CIFAR-100 and 10 in CIFAR-10.

---

[1] https://github.com/sovrasov/flops-counter.pytorch

aL-SAR stores similarities between classes, the training frequency of each sample, and the trace of FIM for each layer. AR needs 400 bytes = 4 bytes/float $\times 10^2$ for saving class-wise similarities, 4 bytes/int $\times |\mathcal{E}|$ for saving frequency of each sample, and 4 bytes/float $\times n_l$ for saving trace of FIM for each layer, where $n_l$ is total number of layers. However, such additional memory cost is negligible compared to episodic memory or model parameters (only 0.1% of memory budget). We summarize implementation details of the total memory budget for each dataset in Tab.17, Tab.18, Tab.19,Tab.20, and Tab.21.

| Methods | $\mathcal{B}$ Type | $S(|\mathcal{B}|)$ | $|\mathcal{E}|$ | $S(|\mathcal{E}|)$ | Model Type | # Parameters | Model Size |
|---|---|---|---|---|---|---|---|
| ER | - | - | 2,000 | 5.85MB | Resnet32 | 0.46M | 1.76MB |
| REMIND | Feature replay | 5.85MB | - | - | Resnet32 | 0.46M | 1.76MB |
| DER | Logits | 0.08MB | 1,974 | 5.77MB | Resnet32 | 0.46M | 1.76MB |
| ER-MIR | - | - | 2,000 | 5.85MB | Resnet32 | 0.46M | 1.76MB |
| EWC | FI & Previous Model | 3.52MB | 794 | 2.33MB | Resnet32 | 0.46M | 1.76MB |
| OCS | - | - | 2,000 | 5.85MB | Resnet32 | 0.46M | 1.76MB |
| X-DER | Logits | 0.08MB | 1,974 | 5.77MB | Resnet32 | 0.46M | 1.76MB |
| LiDER | Logits | 0.08MB | 1,974 | 5.77MB | Resnet32 | 0.46M | 1.76MB |
| MEMO | Expanded Network | 3.52MB | 794 | 2.33MB | Resnet32 | 0.46M | 1.76MB |
| CAMA | Logits | 0.08MB | 1,974 | 5.77MB | Resnet32 | 0.46M | 1.76MB |
| CCL-DC | Teacher model for collaborative learning | 1.76MB | 1397 | 4.08MB | Resnet32 | 0.46M | 1.76MB |
| aL-SAR | Class-wise similarity & frequency of each sample | 8.52KB | 1,997 | 5.84MB | Resnet32 | 0.46M | 1.76MB |

Table 17: Implementation details of total memory budget=7.6MB in CIFAR-10

| Methods | $\mathcal{B}$ Type | $S(|\mathcal{B}|)$ | $|\mathcal{E}|$ | $S(|\mathcal{E}|)$ | Model Type | # Parameters | Model Size |
|---|---|---|---|---|---|---|---|
| ER | - | - | 2,000 | 5.85MB | Resnet32 | 0.46M | 1.76MB |
| REMIND | Feature replay | 5.85MB | - | - | Resnet32 | 0.46M | 1.76MB |
| DER | Logits | 0.71MB | 1,770 | 5.14MB | Resnet32 | 0.46M | 1.76MB |
| ER-MIR | - | - | 2,000 | 5.85MB | Resnet32 | 0.46M | 1.76MB |
| EWC | FI & Previous Model | 3.52MB | 794 | 2.33MB | Resnet32 | 0.46M | 1.76MB |
| OCS | - | - | 2,000 | 5.85MB | Resnet32 | 0.46M | 1.76MB |
| X-DER | Logits | 0.71MB | 1,770 | 5.14MB | Resnet32 | 0.46M | 1.76MB |
| LiDER | Logits | 0.71MB | 1,770 | 5.14MB | Resnet32 | 0.46M | 1.76MB |
| MEMO | Expanded Network | 1.33MB | 1,542 | 4.51MB | Resnet32 | 0.46M | 1.76MB |
| CAMA | Logits | 0.71MB | 1,770 | 5.14MB | Resnet32 | 0.46M | 1.76MB |
| CCL-DC | Teacher model for collaborative learning | 1.76MB | 1397 | 4.08MB | Resnet32 | 0.46M | 1.76MB |
| aL-SAR | Class-wise similarity & frequency of each sample | 0.05MB | 1,987 | 5.83MB | Resnet32 | 0.46M | 1.76MB |

Table 18: Implementation details of total memory budget=7.6MB in CIFAR-100

| Methods | $\mathcal{B}$ Type | $S(|\mathcal{B}|)$ | $|\mathcal{E}|$ | $S(|\mathcal{E}|)$ | Model Type | # Parameters | Model Size |
|---|---|---|---|---|---|---|---|
| ER | - | - | 4,000 | 574.0MB | Resnet32 | 0.46M | 1.76MB |
| REMIND | Feature replay | 574.0MB | - | - | Resnet32 | 0.46M | 1.76MB |
| DER | Logits | 0.1MB | 3,999 | 573.9MB | Resnet32 | 0.46M | 1.76MB |
| ER-MIR | - | - | 4,000 | 574.0MB | Resnet32 | 0.46M | 1.76MB |
| EWC | FI & Previous Model | 4.0MB | 3,972 | 570.0MB | Resnet32 | 0.46M | 1.76MB |
| OCS | - | - | 4,000 | 574.0MB | Resnet32 | 0.46M | 1.76MB |
| X-DER | Logits | 0.1MB | 3,999 | 573.9MB | Resnet32 | 0.46M | 1.76MB |
| LiDER | Logits | 0.1MB | 3,999 | 573.9MB | Resnet32 | 0.46M | 1.76MB |
| MEMO | Expanded Network | 1.4MB | 3,990 | 572.6MB | Resnet32 | 0.46M | 1.76MB |
| CAMA | Logits | 0.1MB | 3,999 | 573.9MB | Resnet32 | 0.46M | 1.76MB |
| CCL-DC | Teacher model for collaborative learning | 1.76MB | 3,987 | 572.2MB | Resnet32 | 0.46M | 1.76MB |
| aL-SAR | Class-wise similarity & frequency of each sample | 0.1MB | 3,999 | 573.9MB | Resnet32 | 0.46M | 1.76MB |

Table 19: Implementation details of total memory budget=574.4MB in CLEAR-10

| Methods | $\mathcal{B}$ Type | $S(|\mathcal{B}|)$ | $|\mathcal{E}|$ | $S(|\mathcal{E}|)$ | Model Type | # Parameters | Model Size |
|---------|--------------------|--------------------|------------------|---------------------|------------|--------------|------------|
| ER | - | - | 8,000 | 1,148.0MB | Resnet32 | 0.46M | 1.76MB |
| REMIND | Feature replay | 1,148.0MB | - | - | Resnet32 | 0.46M | 1.76MB |
| DER | Logits | 3.2MB | 7,978 | 1,144.8MB | Resnet32 | 0.46M | 1.76MB |
| ER-MIR | - | - | 8,000 | 1,148.0MB | Resnet32 | 0.46M | 1.76MB |
| EWC | FI & Previous Model | 4.0MB | 7,972 | 1,144.0MB | Resnet32 | 0.46M | 1.76MB |
| OCS | - | - | 8,000 | 1,148.0MB | Resnet32 | 0.46M | 1.76MB |
| X-DER | Logits | 3.2MB | 7,978 | 1,144.8MB | Resnet32 | 0.46M | 1.76MB |
| LiDER | Logits | 3.2MB | 7,978 | 1,144.8MB | Resnet32 | 0.46M | 1.76MB |
| MEMO | Expanded Network | 1.4MB | 7,990 | 1,146.6MB | Resnet32 | 0.46M | 1.76MB |
| CAMA | Logits | 3.2MB | 7,978 | 1,144.8MB | Resnet32 | 0.46M | 1.76MB |
| CCL-DC | Teacher model for collaborative learning | 1.76MB | 7,987 | 1146.2MB | Resnet32 | 0.46M | 1.76MB |
| aL-SAR | Class-wise similarity & frequency of each sample | 0.1MB | 7,999 | 1,147.9MB | Resnet32 | 0.46M | 1.76MB |

Table 20: Implementation details of total memory budget=1149.8MB in CLEAR-100

| Methods | $\mathcal{B}$ Type | $S(|\mathcal{B}|)$ | $|\mathcal{E}|$ | $S(|\mathcal{E}|)$ | Model Type | # Parameters | Model Size |
|---------|--------------------|--------------------|------------------|---------------------|------------|--------------|------------|
| ER | - | - | 40,000 | 5,740.0MB | Resnet18 | 11.17M | 42.6MB |
| REMIND | Feature replay | 5,740.0MB | - | - | Resnet18 | 11.17M | 42.6MB |
| DER | Logits | 148.6MB | 38,964 | 5,591.4MB | Resnet18 | 11.17M | 42.6MB |
| ER-MIR | - | - | 40,000 | 5,740.0MB | Resnet18 | 11.17M | 42.6MB |
| EWC | FI & Previous Model | 85.2MB | 39,406 | 5,654.8MB | Resnet18 | 11.17M | 42.6MB |
| OCS | - | - | 40,000 | 5,740.0MB | Resnet18 | 11.17M | 42.6MB |
| X-DER | Logits | 148.6MB | 38,964 | 5,591.4MB | Resnet18 | 11.17M | 42.6MB |
| LiDER | Logits | 148.6MB | 38,964 | 5,591.4MB | Resnet18 | 11.17M | 42.6MB |
| MEMO | Expanded Network | 32.0MB | 39,777 | 5,708.0MB | Resnet18 | 11.17M | 42.6MB |
| CCL-DC | Teacher model for collaborative learning | 42.6MB | 39,703 | 5,697.4MB | Resnet18 | 11.17M | 42.6MB |
| aL-SAR | Class-wise similarity & frequency of each sample | 3.9MB | 39,973 | 5,736.1MB | Resnet18 | 11.17M | 42.6MB |

Table 21: Implementation details of total memory budget=5,782.6MB in ImageNet

## A.24 DETAILS ABOUT $A_{\text{AUC}}$

Recent studies (Pellegrini et al., 2020; Caccia et al., 2022; Banerjee et al., 2023; Ghunaim et al., 2023) suggest that having good inference performance at any intermediate time points during training is important for CL. To evaluate intermediate performance during training, (Koh et al., 2022) proposed $A_{\text{AUC}}$, which measures the area under the curve of average accuracy. In contrast to $A_{last}$ or $A_{avg}$ which measures performance only at the end of the task (*i.e.*, after sufficient training), $A_{\text{AUC}}$ consistently measures performance over the course of training. If two methods reach the same accuracy at the end of a task, but one method converges faster than the other, their $A_{last}$ and $A_{avg}$ would be equal, but the faster model would show higher $A_{\text{AUC}}$. Thus, how fast the model adapts to the new task is reflected in $A_{\text{AUC}}$.

## A.25 COMPARISON OF FORGETTING

We compare the forgetting of aL-SAR with other baselines on CIFAR-10, CIFAR-100, and ImageNet-1K, and summarize the results in Fig.18, and Tab. 22, respectively. Specifically, in disjoint setup, we report $F_{last}$ (Chaudhry et al., 2018), following Bang et al. (2021); Koh et al. (2022). In the Gaussian setup, however, the continuous shift in data distribution lacks explicit task boundaries, making traditional metrics like $F_{last}$ (Chaudhry et al., 2018) unsuitable. Therefore, we report the Knowledge Loss Ratio (KLR) (Koh et al., 2023), which do not require task boundaries.

As shown in Fig.18, forgetting decreases as training FLOPs increase. We believe this is because the in-

| Methods | $KLR \downarrow$ | TFLOPs $\downarrow$ |
|---------|------------------|---------------------|
| ER | 61.58 | |
| ER-MIR | 61.05 | |
| DER++ | 60.21 | |
| LiDER | 59.21 | 114,014 |
| X-DER | 59.72 | |
| MEMO | 59.25 | |
| CAMA | 60.02 | |
| CCL-DC | 61.26 | |
| aL-SAR (Ours) | 56.28 | 947,128 |

Table 22: Comparison of KLR and KGR on ImageNet-1K Gaussian scheduled setup.

creased computational budget allows the model to sufficiently train on previously encountered data before being exposed to new data.

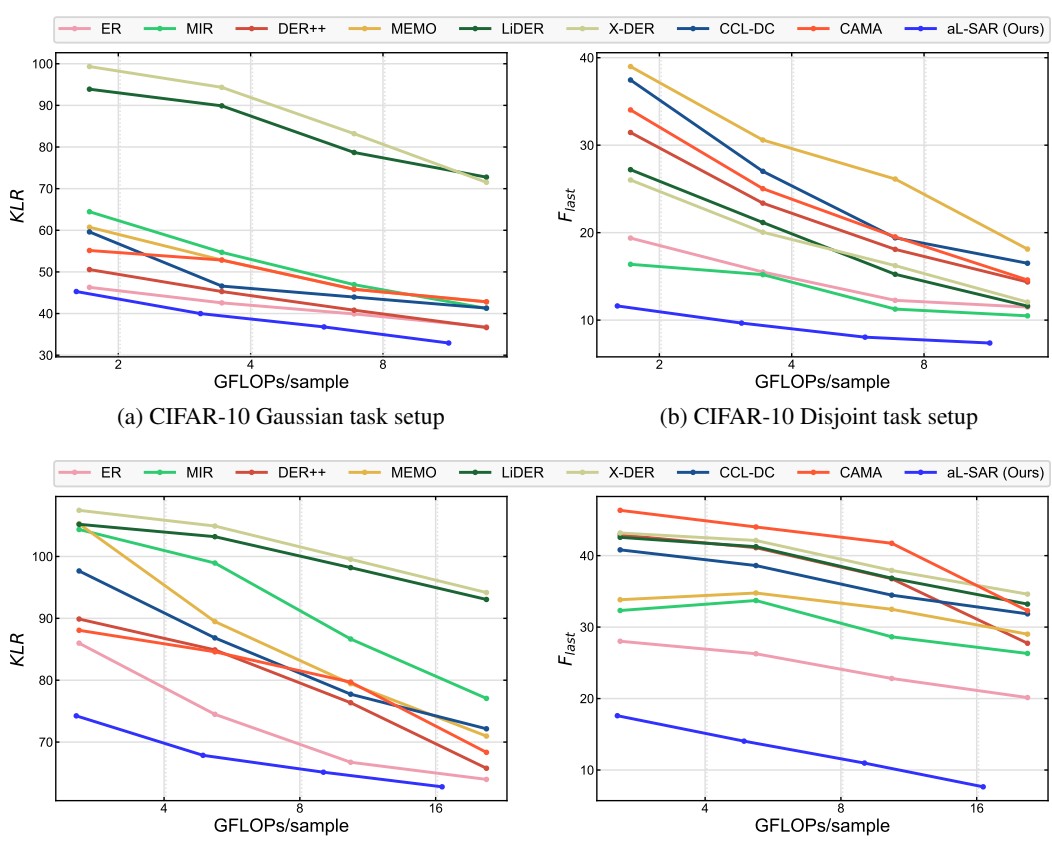

(a) CIFAR-10 Gaussian task setup    (b) CIFAR-10 Disjoint task setup

(a) CIFAR-100 Gaussian task setup    (b) CIFAR-100 Disjoint task setup

Figure 18: Comparison of forgetting on Gaussian and Disjoint CL setup in CIFAR-10 and CIFAR-100.

## A.26 TOY EXPERIMENTS ON SIMILARITY-AWARE RETRIEVAL

We perform several toy experiments to validate the motivations for our proposed retrieval method. To isolate the effect of each sample, we train ResNet-20 from scratch with batch size of 1 using a subset of CIFAR-10. First, to validate our claim *"extensively used samples provide little knowledge to the model"*, we plot (use frequency) vs (training loss decrease caused by training with the sample) in Fig. 19. We observe that training with a frequently used sample results in a small decrease in training loss.

Next, we validate the exponential decaying model of forgetting. To measure this, we first train the model with one sample and measure the use frequency along with the corresponding training loss. Next, when training the model with different samples, we measure the loss of the previously trained sample. At this point, due to the training of other samples, forgetting occurred for the sample we used for initial training, causing the training loss to increase again. We then use the measured use-frequency and corresponding training loss values, which are measured in the first stage, to reverse-transform the training loss into an effective use-frequency. As shown in Fig. 20, as the number of training iterations for other samples increases, we observe that the training loss for previously trained samples increases, leading to a decrease in the effective use-frequency. Note that the $y$-axis is in log scale and the plot shows a linear trend, showing that the loss increase from forgetting corresponds to exponential decay of use frequency.

Finally, we validate that cosine similarity between gradients can be used to measure the effective change in use frequency from using other samples. Using the same setup as the previous experiment, in Fig. 21, we show a scatter plot between gradient similarity and the effective change in use-frequency obtained by the relation between the sample's loss and use frequency. When training

with samples that have high gradient similarity with the reference samples, the effective use frequency of the reference samples increases. On the contrary, when training with samples that have low gradient similarity, the effective use frequency of reference samples decreases. We observe a linear relationship between gradient similarity and the change in effective use-frequency. In other words, the change in effective use frequency can be predicted using gradient similarity.

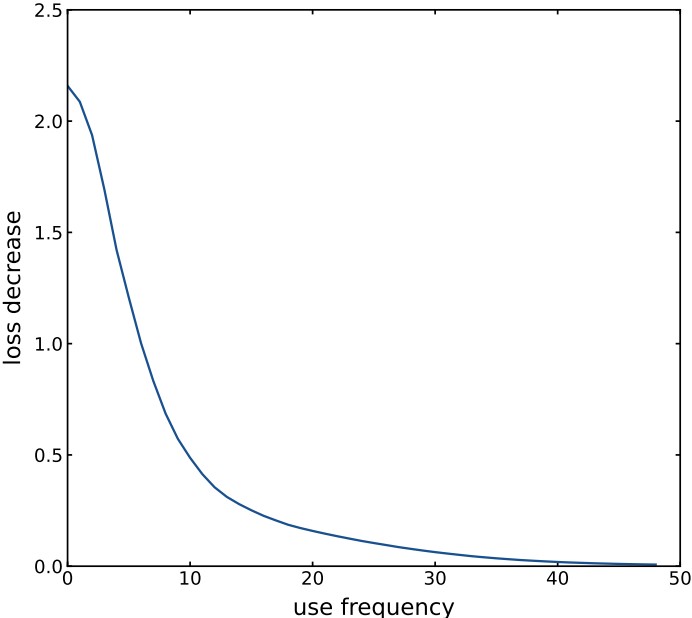

Figure 19: Relation between a sample's use frequency and training loss decrease from training with the sample, measured in CIFAR-10 with ResNet-20. When a sample is used more frequently in training, it has less effect on reducing training loss.

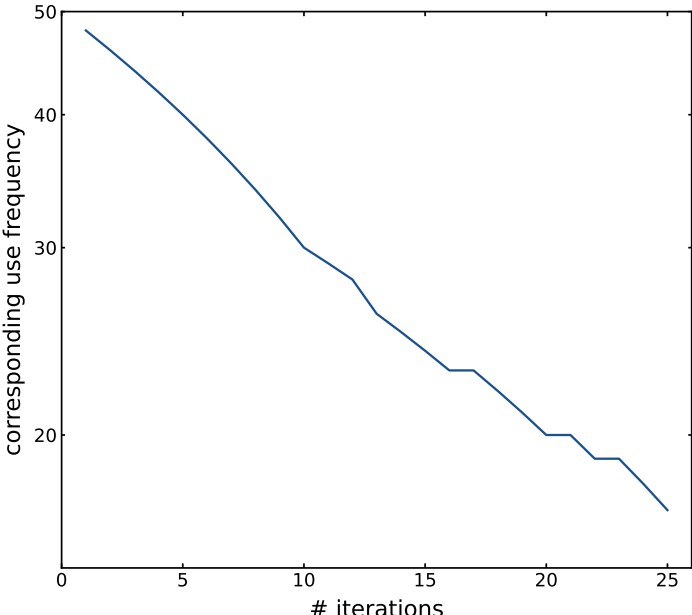

Figure 20: Plot showing the number of iterations from the point where the sample has not been used for training, and the decrease in effective use frequency corresponding to the increase in loss. It is measured in CIFAR-10 with ResNet-20.

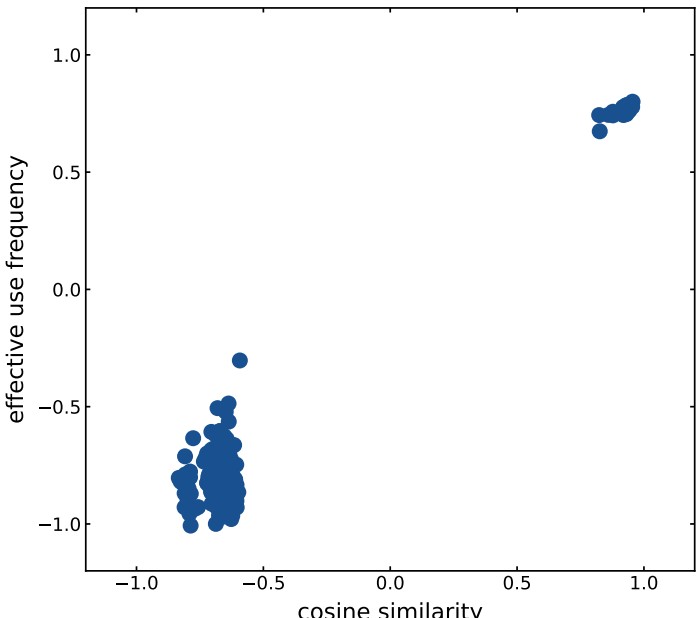

Figure 21: The scatter plot shows the cosine similarity between the gradient of a reference sample and the gradients of other samples, along with the change in the effective use frequency of the reference sample when training with other samples. It is performed in CIFAR-10 with ResNet-20.

### A.27 LIMITATIONS AND FUTURE WORK

While our method only requires negligible additional memory other than episodic memory, it does not actively optimize the memory efficiency of CL algorithms. It is interesting to explore a method to use the limited storage budget more efficiently, *e.g.*, storing quantized versions of models and exemplars.

### A.28 IMPACT STATEMENT

This work aims to update a model in a computationally efficient and online manner without access to training samples used before. Thus, although there is no intent from the authors, this method may exacerbate unsolved issues in deep learning such as model bias and less aligned to ethical standards. We will take all available measures to prevent such outcomes, though that is *not* our intention *at all*.

