# OpenReview forum: "Budgeted Online Continual Learning by Adaptive Layer Freezing and Frequency-based Sampling"
_ICLR.cc/2025/Conference — ICLR 2025 Spotlight_

### Official Review · Reviewer_3EaD · 2024-10-28

**Soundness:** 3
**Presentation:** 3
**Contribution:** 2
**Rating:** 8
**Confidence:** 4

**Summary:**

This paper addresses the challenge of resource constraints in online continual learning (CL) by introducing an efficient method named aL-SAR, which combines adaptive layer freezing and similarity-aware sample retrieval. The adaptive layer freezing component selectively freezes layers to minimize computation based on Fisher Information, while the similarity-aware retrieval component focuses on sampling underused and informative samples from episodic memory to enhance training efficiency. Experiments on CIFAR-10/100, CLEAR-10/100, and ImageNet-1K datasets demonstrate that aL-SAR outperforms several state-of-the-art methods under the same computational and memory budgets.

**Strengths:**

Pros:

1. The paper introduces a unique approach to online CL by integrating adaptive layer freezing and frequency-based sampling, effectively reducing both computational and memory costs without significant performance trade-offs.

2. The empirical results across multiple datasets comparing to multiple existing SOTA benchmarks show the competitiveness of al-SAR, especially under constrained resource settings.

**Weaknesses:**

1. Though designed to be efficient by only computing class based gradient, the similarity-aware retrieval approach still requires gradient similarity calculations, which could introduce overhead in large-scale applications.

2. The experimental setups impose strict memory and computational constraints, limiting the contribution primarily to incremental improvements in the efficiency of existing online continual learning approaches.

3. The authors should compare the method with model based like iCaRL.

**Questions:**

Please also refer the comments in the strengths and weaknesses sections:

1. Can the authors compare the al-SAR with other methods under the epoch-based setup to show the effectiveness of the method with larger memory constraint?

2. Can you consider applying aL-SAR on more diverse datasets (e.g., text, multimodal data) to demonstrate the approach’s generalizability?

---

> ### Author Response · Authors · 2024-11-20
> **Answers to the questions of Reviewer 3EaD (1/3)**
>
> > Though designed to be efficient by only computing class based gradient, the similarity-aware retrieval approach still requires gradient similarity calculations, which could introduce overhead in large-scale applications.
>
> $\to$ We can minimize the computational overhead in large-scale applications by introducing a hierarchical structure that groups similar classes together. This allows us to focus on group-wise similarity rather than class-wise similarity, thereby reducing computational overhead. Since many real-world scenarios involve hierarchical relationships between old and new tasks [1], incorporating class groups enables aL-SAR to better address these real-world challenges. Moreover, as mentioned in Sec.A.17, by utilizing only 0.05% of the entire model’s parameter gradients, we believe that aL-SAR will maintain computational efficiency in large-scale applications.
>
> > The authors should compare the method with model based like iCaRL.
>
> $\to$ Sure. We add comparisons with iCaRL in the table below. In CIFAR-10 experiments, we set the computational budget and memory budget as 293.6 TFLOPs and 25.16 MB, respectively. In CIFAR-100 experiments, we set the computational budget and memory budget as 440.4 TFLOPs and 25.16 MB, respectively. Note that aL-SAR has lower FLOPs by virtue of proposed adaptive layer freezing.
>
> **Table A. Comparison between aL-SAR and iCaRL on CIFAR-10 Gaussian Setup under total constraint**
> | Model | $A_\text{AUC}\uparrow$ | $A_{last}\uparrow$ | TFLOPs $ \downarrow$ |
> |----------------------|---------|---------|------------|
> | iCaRL | 62.03$\pm$2.05 | 77.55$\pm$1.42 | 171.93 |
> | aL-SAR (Ours) | **69.35**$\pm$**1.65** | **82.44**$\pm$**1.92** | **141.46** |
>
> **Table B. Comparison between aL-SAR and iCaRL on CIFAR-10 Disjoint Setup under total constraint**
> | Model | $A_\text{AUC}\uparrow$ | $A_{last}\uparrow$ | TFLOPs $ \downarrow$ |
> |----------------------|---------|---------|------------|
> | iCaRL | 81.62$\pm$0.88 | 78.63$\pm$0.92 | 171.93 |
> | aL-SAR (Ours) | **85.04**$\pm$**0.65** | **82.07**$\pm$**1.13** | **141.46** |
>
> **Table C. Comparison between aL-SAR and iCaRL on CIFAR-100 Gaussian Setup under total constraint**
> | Model | $A_\text{AUC}\uparrow$ | $A_{last}\uparrow$ | TFLOPs $ \downarrow$ |
> |----------------------|---------|---------|------------|
> | iCaRL | 34.00$\pm$0.60 | 44.66$\pm$1.27 | 257.90 |
> | aL-SAR (Ours) | **42.38**$\pm$**1.12** | **50.62**$\pm$**1.40** | **223.95** |
>
> **Table D. Comparison between aL-SAR and iCaRL on CIFAR-100 Disjoint Setup under total constraint**
> | Model | $A_\text{AUC}\uparrow$ | $A_{last}\uparrow$ | TFLOPs $ \downarrow$ |
> |----------------------|---------|---------|------------|
> | iCaRL | 47.14$\pm$1.01 | 45.54$\pm$0.73 | 257.90 |
> | aL-SAR (Ours) | **53.33**$\pm$**1.33** | **47.14**$\pm$**1.25** | **222.83** |

---

> ### Author Response · Authors · 2024-11-20
> **Answers to the questions of Reviewer 3EaD (2/3)**
>
> > Can the authors compare the al-SAR with other methods under the epoch-based setup to show the effectiveness of the method with larger memory constraint?
>
> $\to$ Yes, we compare aL-SAR with baselines under the epoch-based constraint and a large memory constraint, summarizing the results in the table below. Specifically, we assume a memory infinite setup [2], which allows storing all samples in episodic memory, as well as logits and previous models for distillation.
>
> While ER outperforms other methods except our proposed aL-SAR in total constraint, ER-CCLDC and MIR outperforms baselines. However, MIR and ER-CCLDC consumes 4.12 times and 4.66 times more FLOPs than ER, as MIR requires additional forward and backward passes to retrieve maximally interfered samples from episodic memory, while ER-CCLDC requires 10 times more forward passes for collaborative learning and distillation using augmentations. Additionally, ER-CCLDC requires a sibling network for collaborative learning, which further increases memory storage requirements. In contrast, aL-SAR outperforms baselines with less computational cost and negligible additional memory budget.
>
> As discussed in Line 39 and Fig. 1, we argue that comparing methods under the same epoch constraint does not provide useful information about computational cost as a method can do a lot of computation during a single epoch which may take much longer time than other methods. Furthermore, we would like to emphasize that we compare aL-SAR with baselines under the large memory constraint (i.e., memory infinite setup) in Sec. A.21, where aL-SAR demonstrates superior performance compared to the memory constraint.
>
> **Table E. Comparison between aL-SAR and other baselines on the CIFAR-100 Gaussian Setup under total constraints, using 3 epochs of training per sample encounter with an infinite memory budget.**
> | Model | $A_\text{AUC}\uparrow$ | $A_{last}\uparrow$ | $TFLOPs \uparrow$ | Memory (MB) $\downarrow$ |
> |----------------------|---------|---------|---------|---------|
> | ER | 36.52$\pm$0.57 | 47.27$\pm$1.90  | 518.44 | **148.01** |
> | DER | 31.07$\pm$0.70 | 36.69$\pm$1.32 | 518.44 | 168.45 |
> | MIR | 38.89$\pm$0.64 | 50.82$\pm$0.64 | 2,135.97 | **148.01** |
> | OCS | 37.33$\pm$1.93 | 48.43$\pm$2.07 | 1,036.88 | **148.01** |
> | XDER | 29.90$\pm$0.64 | 37.68$\pm$1.22 | 1207.96 | 168.45 |
> | MEMO | 31.49$\pm$0.50 | 34.10$\pm$2.61 | 772.48 | 149.34 |
> | CAMA | 27.96$\pm$1.02 | 31.38$\pm$1.47 | 518.44 | 168.45 |
> | CCL-DC | 40.78$\pm$0.52 | 48.89$\pm$0.93 | 2415.93 | 149.77 |
> | aL-SAR (Ours) | **42.50**$\pm$**0.51** | **56.78**$\pm$**0.40** | **472.92** | **148.01** |
>
>
> **Table F. Comparison between aL-SAR and other baselines on the CIFAR-100 Disjoint Setup under total constraints, using 2 epochs of training per sample encounter with an infinite memory budget.**
> | Model | $A_\text{AUC}\uparrow$ | $A_{last}\uparrow$ | $TFLOPs \uparrow$ | Memory (MB) $\downarrow$ |
> |----------------------|------------|------------|------------|------------|
> | ER | 49.87$\pm$1.02 | 48.82$\pm$0.38 | 518.44 | **148.01** |
> | DER | 40.69$\pm$0.45 | 37.06$\pm$0.15 | 518.44 | 168.45 |
> | MIR | 52.33$\pm$1.13 | 50.82$\pm$0.16 | 2,135.97 | **148.01** |
> | OCS | 33.90$\pm$2.51 | 44.29$\pm$0.56 | 1,036.88 | **148.01** |
> | XDER | 38.63$\pm$0.83 | 35.73$\pm$0.18 | 1207.96 | 168.45 |
> | MEMO | 39.71$\pm$0.91 | 32.84$\pm$1.01 | 772.48 | 149.34 |
> | CAMA | 36.12$\pm$0.84 | 31.22$\pm$0.52 | 518.44 | 168.45 |
> | CCL-DC | 53.61$\pm$1.28 | 48.55$\pm$0.78 | 2415.93 | 149.77 |
> | aL-SAR (Ours) | **54.62**$\pm$**1.38** | **56.52**$\pm$**0.98** | **458.63** | **148.01** |
>
>
> **Table G. Comparison between aL-SAR and other baselines on the ImageNet Gaussian Setup under total constraints, using 0.125 epochs of training per sample encounter with an infinite memory budget.**
> | Model | $A_\text{AUC}\uparrow$ | $A_{last}\uparrow$ | $PFLOPs \downarrow$ | Memory (GB) $\downarrow$ |
> |----------------------|-----------|-----------|-----------|-----------|
> | ER | 22.89 | 38.44 | 11.40 | **176.67** |
> | DER | 23.53 | 30.36 | 11.40 | 209.39 |
> | MIR | 26.75 | 42.05 | 46.97 | **176.67** |
> | OCS | 22.91 | 30.57 | 22.80 | **176.67** |
> | XDER | 24.30 | 33.50 | 26.56 | 209.39 |
> | MEMO | 13.14 | 17.05 | 16.99 | **176.67** |
> | CAMA | 24.62 | 30.72 | 11.40 | 209.39 |
> | CCL-DC | 30.54 | 44.06 | 53.12 | **176.67** |
> | aL-SAR (Ours) | **37.89** | **52.47** | **9.92** | **176.67** |

---

> ### Author Response · Authors · 2024-11-20
> **Answers to the questions of Reviewer 3EaD (3/3)**
>
> > Can you consider applying aL-SAR on more diverse datasets (e.g., text, multimodal data) to demonstrate the approach’s generalizability?
>
> $\to$ As suggested, we applied the proposed aL-SAR to a multi-modal concept-incremental task, using LLaVA-1.5-7B as the architecture and multi-modal data such as Bongard-OpenWorld and Bongard-HOI (details are provided in Sec. A.13), and we summarized the results in Tables H and I. We compare aL-SAR with ER, the best-performing baseline, under the total constraint.
>
> **Table H. Comparison between aL-SAR and ER on the Bongard-HOI under total constraint (Memory budget : 27.05GB)**
> | Model | $A_\text{AUC}\uparrow$ | $A_{last}\uparrow$ | TFLOPs $\downarrow$ |
> |----------------------|---------|---------|---------|
> | ER | 59.85$\pm$1.55 | 49.35$\pm$1.76 | 1578.02 |
> | aL-SAR (Ours) | **62.34**$\pm$**1.54** | **52.77**$\pm$**2.82** | **1418.82** |
>
>
> **Table I. Comparison between aL-SAR and ER on the Bongard-OpenWorld under total constraint (Memory budget : 27.05GB)**
> | Model | $A_\text{AUC}\uparrow$ | $A_{last}\uparrow$ | TFLOPs $\downarrow$ |
> |----------------------|---------|---------|---------|
> | ER | 55.95$\pm$1.71 | 58.56$\pm$2.48 | 2959.61 |
> | aL-SAR (Ours) | **59.36**$\pm$**2.56** | **60.13**$\pm$**2.25** | **2619.83** |
>
> [1] Lee et al., Online Continual Learning on Hierarchical Label Expansion, ICCV 2023
>
> [2] Prabhu et al., Online Continual Learning Without the Storage Constraint, arXiv 2023

---

> ### Author Response · Authors · 2024-11-25
> **Discussion Reminder**
>
> We sincerely appreciate your effort in reviewing our submission. We gently remind the reviewer that we tried our best to address your concerns via our replies and manuscript revision. As the discussion period is nearing its end, we would be glad to hear more from you if there are further concerns.

---

> > ### Comment · Reviewer_3EaD · 2024-11-26
> > **Response to rebuttal**
> >
> > I do not have further questions, the response has addressed my concerns well, I will increase my score to 8

---

> > > ### Author Response · Authors · 2024-11-27
> > > **Thank you for your response**
> > >
> > > Dear Reviewer 3EaD,
> > >
> > > Thank you for your comments and for taking the time to review our manuscript.
> > >
> > > We are happy that our additional discussion and experiments addressed your concerns!
> > >
> > > If you have any further comments or suggestions, please let us know. We are committed to improving the quality of our work, and we value your feedback.
> > >
> > > Thank you very much,
> > >
> > > Authors

---

### Official Review · Reviewer_VVoh · 2024-11-01

**Soundness:** 3
**Presentation:** 3
**Contribution:** 3
**Rating:** 6
**Confidence:** 3

**Summary:**

This paper addresses the challenge of fair comparison in online continual learning (CL) by proposing a total resource budget framework that considers both computational costs (FLOPs) and memory usage (bytes). The authors introduce aL-SAR, which combines two key innovations:

1. Adaptive layer freezing: A method that selectively freezes network layers based on Fisher Information to reduce computational costs while maintaining performance
2. Similarity-aware retrieval: A computationally efficient strategy for retrieving training samples based on usage frequency and gradient similarity

The method is extensively validated on CIFAR-10/100, CLEAR-10/100, and ImageNet-1K datasets, showing superior performance compared to state-of-the-art methods under the same resource constraints.

**Strengths:**

1. Novel framework for comparing CL methods using total resource budgets including innovative adaptive layer freezing approach based on Fisher Information and computationally efficient memory retrieval strategy.

2. Rigorous empirical validation across multiple datasets and setups containing thorough ablation studies and comparisons with state-of-the-art methods and successful application to large models (LLaVA-1.5-7B).

3. Addresses practical constraints in online CL and shows significant performance improvements while reducing computational costs.

**Weaknesses:**

1. Using the layer freezing technque for recuding the model params would always block the shallow layers (i.e., layer 1-n) and their representation would be outdated. This would be problematic for frequent distribution shifts.

2. The paper's adaptive layer freezing strategy treats layers somewhat independently when making freezing decision. However, neural networks typically have complex inter-layer relationships that this approach might not fully capture.

3. The method doesn't address catastrophic forgetting directly, instead focusing on resource efficiency. It's unclear how the method balances the trade-off between resource efficiency and forgetting prevention.

4. The computational overhead of maintaining class-wise gradient similarities might become prohibitive for problems with a large number of classes or frequent distribution shifts.

**Questions:**

1. Could the adaptive layer freezing strategy be extended to handle cross-layer dependencies, potentially leading to more optimal freezing decisions?
2. Although authors have illustrated the process of calculating *Effective Use Frequency* in Sec.3.2 and Sec.A.17, it is still vague for me that how to calculate cosine similarity of the gradients between classes? In A.17, do it that using *0.05% of the model parameters* means that randomly sample params among unfrozen layers?

---

> ### Author Response · Authors · 2024-11-20
> **Answers to the questions of Reviewer VVoh (1/2)**
>
> > Using the layer freezing technque for recuding the model params would always block the shallow layers (i.e., layer 1-n) and their representation would be outdated. This would be problematic for frequent distribution shifts.
>
> $\to$ Thank you for the insightful comments. But our proposed adaptive layer freezing method (i.e., aL) does not encounter such issues during frequent distribution shifts, since when temporal distribution shifts occur, the adaptive layer freezing (called ‘aL’) tends not to freeze layers, including shallow ones. Specifically, as shown in Fig.8 in Sec.A.10, we observe that batch freezing criterion (BFC) values, which quantify the net benefit in the information gained by the model when we freeze layers given an input batch $z_t$, tend to remain negative on average, even in shallow layers (note that the plot is significantly smoothed to highlight overall BFC trends, while actual values fluctuate depending on the input batch). This indicates that freezing is not beneficial for most batches, likely due to insufficient training in continual learning and the continuous introduction of new samples. However, for input batches with little information, BFC temporarily becomes positive, allowing for freezing.
>
> Furthermore, at the task boundaries, we observe a sharp drop in BFC values, indicating that the layers are less likely to be frozen, allowing them to learn from the new, previously unseen data. This demonstrates that BFC effectively handles distribution shifts, even though our method does not rely on any task boundary information. Additionally, while shallow layers exhibit higher BFC values compared to deeper layers, suggesting they are more frequently eligible for freezing, they are not always frozen, especially when faced with shifted distributions.
>
> Moreover, we demonstrate that aL-SAR performs effectively under frequent distribution shifts by conducting experiments in the Gaussian-scheduled setup, where the data distribution follows a Gaussian distribution and shifts at each time step, unlike the disjoint setup, which assumes distribution changes only at explicit task boundaries.
>
> > Although authors have illustrated the process of calculating Effective Use Frequency in Sec.3.2 and Sec.A.17, it is still vague for me that how to calculate cosine similarity of the gradients between classes? In A.17, do it that using 0.05% of the model parameters means that randomly sample params among unfrozen layers?
>
> $\to$ No, not just among the unfrozen layers but **across all layers before training**. Specifically, we first randomly select 0.05% of the model parameters. Then, for each iteration, we use the gradients of the pre-selected, which are unfrozen by adaptive layer freezing, to update the class-wise gradients. Specifically, we employ an EMA estimation of the class-wise gradients, updating them every batch iteration using the gradients of selected parameters in an EMA manner. While not all EMA-estimated class-wise gradients are updated every iteration (since gradients of frozen layers are not computed), all elements of the estimated class-wise gradients are updated every $m$ iterations, as we unfreeze all layers every $m$ iterations, as mentioned in Sec. A.1. To form the similarity matrix, we first compute the element-wise product of class-wise gradients and then average them over all class-wise pairs.
>
> Note that the reason for using only 0.05% of the model parameters for the similarity calculation is to reduce the computational cost, as the gradient distribution of this randomly selected subset is similar to that of the entire weight set [1]. We have added these details in Sec. A.17 in the revision for further clarification. Thank you!
>
> > The paper's adaptive layer freezing strategy treats layers somewhat independently when making freezing decision. However, neural networks typically have complex inter-layer relationships that this approach might not fully capture. Could the adaptive layer freezing strategy be extended to handle cross-layer dependencies, potentially leading to more optimal freezing decisions?
>
> $\to$ This is a terrific point. Yes, we treat the layers independently but we can extend it to account for cross-layer dependencies by incorporating non-diagonal elements of the Fisher Information Matrix (FIM), which involves computing the Hessian (i.e., second-order derivatives). However, given our goal is **computationally efficient CL**, we intentionally limit our focus to the diagonal components of the FIM (i.e., treating each layer independently). Note that calculating these diagonal components only requires first-order derivatives, which are naturally obtained during gradient descent. Considering layer-combinatorial methods that are computationally efficient is a very promising future research avenue.

---

> ### Author Response · Authors · 2024-11-20
> **Answers to the questions of Reviewer VVoh (2/2)**
>
> > The method doesn't address catastrophic forgetting directly, instead focusing on resource efficiency. It's unclear how the method balances the trade-off between resource efficiency and forgetting prevention.
>
> $\to$ Thank you for your insightful question. The proposed adaptive layer freezing **inherently (or naturally) balances** the trade-off by preventing unnecessary parameter overwriting. In other words, our proposed adaptive layer freezing provides advantages in both resource efficiency and forgetting prevention. Specifically, we believe this advantage stems from preventing parameter overwriting, which is one of the primary causes of catastrophic forgetting [4, 5]. Specifically, when model parameters are continually overwritten by new data, fully updating the model at every iteration can result in severe forgetting. On the other hand, freezing parameters can limit the model’s plasticity to learn new concepts [6]. To address this stability-plasticity dilemma, our proposed aL adaptively freezes layers based on the informativeness of each training batch, updating parameters only when the batch contains sufficiently informative data. Since each training batch includes both recently encountered and past data from episodic memory, aL freezing may freeze fewer layers for batches containing undertrained or forgotten data, while freezing more layers for well-trained batches. This approach prevents unnecessary parameter overwriting and maintains a balance between stability and plasticity.
>
> Furthermore, we account for forgetting in similarity-aware retrieval (SAR) by using discounted use frequency, as described in Sec. 3.2. If a sample was frequently used in the past but seldom used in recent iterations, its knowledge may have been forgotten despite its high use frequency. To account for this, we apply an exponential decay to the use frequency, inspired by the exponential decay model of forgetting.
> To demonstrate the effectiveness of forgetting prevention in aL-SAR, we include a comparison of forgetting rates between aL-SAR and the baselines in Sec. A.24, showing reduced forgetting across various benchmarks and setups.
>
> > The computational overhead of maintaining class-wise gradient similarities might become prohibitive for problems with a large number of classes or frequent distribution shifts.
>
> $\to$ We can minimize the computational overhead in large-scale applications by introducing a hierarchical structure that groups similar classes together. This allows us to focus on group-wise similarity rather than class-wise similarity, thereby reducing computational overhead. Since many real-world scenarios involve hierarchical relationships between old and new tasks [7], incorporating class groups enables aL-SAR to better address these real-world challenges. Moreover, as mentioned in Sec.A.17, by utilizing only 0.05% of the entire model’s parameter gradients, we believe that aL-SAR will maintain computational efficiency in large-scale applications.
>
> [1] Li et al., Smartfrz: An efficient training 415 framework using attention-based layer freezing, ICLR 2023
>
> [2] Yuan et al., Layer Freezing & Data Sieving: Missing Pieces of a Generic Framework for Sparse Training, NeurIPS 2022
>
> [3] Liu et al., AutoFreeze: Automatically Freezing Model Blocks to Accelerate Fine-tuning, arXiv 2021
>
> [4] Zhang et al., Continual Learning on Dynamic Graphs via Parameter Isolation, SIGIR 2023
>
> [5] Kirkpatricka et al., Overcoming catastrophic forgetting in neural networks,PNAS 2017
>
> [6] Zhao et al.,SAFE: Sl ow and Fast Parameter-Efficient Tuning for Continual Learning with Pre-Trained Models, NeurIPS 2024
>
> [7] Lee et al., Online Continual Learning on Hierarchical Label Expansion, ICCV 2023
>
> [8] Lee et al., Layer-Wise Adaptive Model Aggregation for Scalable Federated Learning, AAAI 2023
>
> [9] Yuan et al., Layer Freezing & Data Sieving: Missing Pieces of a Generic Framework for Sparse Training, NeurIPS 2022

---

> ### Author Response · Authors · 2024-11-25
> **Discussion Reminder**
>
> We sincerely appreciate your effort in reviewing our submission. We gently remind the reviewer that we tried our best to address your concerns via our replies and manuscript revision. As the discussion period is nearing its end, we would be glad to hear more from you if there are further concerns.

---

> ### Author Response · Authors · 2024-11-28
> **Discussion Reminder (closing in a few days)**
>
> Dear Reviewer VVoh
>
> We sincerely appreciate your effort in reviewing our submission. We gently remind the reviewer that we tried our best to address your concerns via our replies and manuscript revision. As the discussion period is nearing its end (closing in a few days), we would be glad to hear more from you if there are further concerns.

---

> > ### Comment · Reviewer_VVoh · 2024-12-02
> > **Thanks**
> >
> > I havre read your responses and would like to keep my score. Thanks.

---

> > > ### Author Response · Authors · 2024-12-02
> > > **Official Comment by Authors**
> > >
> > > Thank you very much for your response. Please let us know any of your remaining concerns. We are fully committed to providing further clarifications or conducting additional experiments to address any remaining concerns and further enhance our work.

---

> ### Author Response · Authors · 2024-12-01
> **Discussion Reminder (closing in a few days)**
>
> Dear Reviewer VVoh
>
> We sincerely appreciate your effort in reviewing our submission. We gently remind the reviewer that we tried our best to address your concerns via our replies and manuscript revision. As the discussion period is nearing its end (closing in a few days), we would be glad to hear more from you if there are further concerns.

---

### Official Review · Reviewer_j3Dm · 2024-11-04

**Soundness:** 3
**Presentation:** 3
**Contribution:** 3
**Rating:** 8
**Confidence:** 3

**Summary:**

This paper proposes a new algorithm for the online CL problem. The authors first explain that a fair comparison between different methods should consider the total number of FLOPs instead of the total number of iterations, as some methods can perform expensive operations each round. Also, it is essential to include all types of memory usage rather than only considering the number of examples in the episodic memory as the actual memory cost. Next they propose an efficient algorithm to address the problems in online continual learning. They use batch fisher information to find the optimal layers that can be frozen while learning from the data. Then, they introduce a new metric, 'appearance frequency,' to measure the contributions of each training example.

**Strengths:**

* The arguments about the FLOPS and memory costs are critical. To have a fair and meaningful comparison between different methods, we *need* to know the actual cost.

* The idea of having batch-wise and dynamic layer freezing makes the training more suitable for online settings.

* The authors did an excellent job of explaining the motivation and algorithm. All the different components are adequately explained and justified.

**Weaknesses:**

* In my opinion, the introduction lacks proper background knowledge, and it jumps to the solutions for the existing problems in online CL. I suggest adding at least one paragraph explaining the background and current state of online continual learning and the current high-level challenges that prior work has attempted to solve, so the readers are on the same page and maybe appreciate your work even more.

* Have you considered reporting forgetting as an additional metric? This is commonly used in CL papers and could provide further insight into the efficacy of your method.

**Questions:**

* What is the memory selection mechanism? In other words, how memory samples are selected, and how does the memory get updated?

* Besides the similarity-aware retrieval, is there any other difference between the in-memory and new samples in the training?

---

> ### Author Response · Authors · 2024-11-20
> **Answers to the questions of Reviewer j3Dm**
>
> > In my opinion, the introduction lacks proper background knowledge, and it jumps to the solutions for the existing problems in online CL. I suggest adding at least one paragraph explaining the background and current state of online continual learning and the current high-level challenges that prior work has attempted to solve
>
> $\to$ Thank you for the suggestion! We add a paragraph explaining the background and the current state of online CL at the beginning of the revision as suggested. We highlight why research on online CL has recently garnered interest, discusses various recently proposed methods, and outlines the goals and challenges associated with online CL.
>
> > Have you considered reporting forgetting as an additional metric? This is commonly used in CL papers and could provide further insight into the efficacy of your method.
>
> $\to$ Great suggestion! In Sec. A.24 of the revision, we add a comparison of forgetting between our proposed aL-SAR and baseline methods as suggested. For the disjoint setup, which has clear task boundaries, we report traditional forgetting [1]. In contrast, for the Gaussian-scheduled setup—where explicit task boundaries are absent due to a continuous shift in the data distribution modeled by a Gaussian distribution—we report the Knowledge Loss Ratio (KLR) and Knowledge Gain Ratio (KGR) [2], as they do not require explicit task boundaries.
>
> > What is the memory selection mechanism? In other words, how memory samples are selected, and how does the memory get updated?
>
> $\to$ We use Greedy Balanced Sampling [3] for memory selection strategy (L831).
>
> > Besides the similarity-aware retrieval, is there any other difference between the in-memory and new samples in the training?
>
> $\to$ No, there is no difference. Since we compose mini-batch by retrieving samples only from memory (*i.e.*, memory only training), we do not differentiate between in-memory samples and new samples.  However, since new samples may have been used less frequently, they might have a higher probability of being selected.
>
> [1] Chaudhry et al., Riemannian walk for incremental learning: Understanding forgetting and intransigence, ECCV 2018
>
> [2] Koh et al., Online Boundary-Free Continual Learning by Scheduled Data Prior, ICLR 2023
>
> [3] Prabhu et al., GDumb: A Simple Approach that Questions Our Progress in Continual Learning, ECCV 2020

---

> > ### Comment · Reviewer_j3Dm · 2024-11-24
> > **Response to rebuttal**
> >
> > I do not have any further questions, and after reading other reviews and authors' responses, I have decided to increase my score to **8**.

---

> > > ### Author Response · Authors · 2024-11-25
> > > **Thank you for your response**
> > >
> > > Dear Reviewer j3Dm,
> > >
> > > Thank you for your comments and for taking the time to review our manuscript.
> > >
> > > We are happy that our additional discussion and experiments addressed your concerns!
> > >
> > > If you have any further comments or suggestions, please let us know. We are committed to improving the quality of our work, and we value your feedback.
> > >
> > > Thank you very much,
> > >
> > > Authors

---

### Official Review · Reviewer_1YwR · 2024-11-04

**Soundness:** 3
**Presentation:** 3
**Contribution:** 3
**Rating:** 8
**Confidence:** 4

**Summary:**

This paper proposes the method *aL-SAR*, which standardizes computational and memory budgets of continual learning (CL) algorithms based on training floating point operations (FLOPs) and total memory size in Byte to address the challenge of fair comparison in online CL algorithms. Besides, this paper introduces adaptive layer freezing to reduce computation costs by selectively freezing layers and frequency-based sampling to prioritize under-learned samples. Experiments on CIFAR-10/100 and ImageNet-1K datasets show that aL-SAR consistently outperforms existing CL methods like ER (Experience Replay), DER++, MIR (Maximally Interfered Retrieval), MEMO, REMIND, EWC, OCS, LiDER, X-DER, CCL-DC, and CAMA by up to 5-10% on average across AAUC and last accuracy, especially under stringent budget constraints.

**Strengths:**

++ Standardizing computational and memory budgets of continual learning (CL) algorithms is important for evaluating algorithm efficiency and learning system design.

++ The paper provides comprehensive and detailed experimental results.

**Weaknesses:**

-- Using FLOPs as a metric focuses on raw operations, ignoring algorithm-specific optimizations for training software and hardware systems. See C1.

-- The additional computations required to decide which layers to freeze may introduce overhead that is not directly part of the core training but still contributes to the total computational cost. See C2.

-- The retrieval strategy based on the use-frequency cannot ensure that the model does learn sufficiently by preferring less frequently used samples. See C3.

**Questions:**

C1:
  - FLOPs served as a direct measure of computational demand across different models, making it possible to compare CL algorithms fairly. However, it doesn’t account for real-world runtime or memory access patterns, which vary based on hardware and software implementations. For instance, two architectures with similar FLOPs might have different inference times on the same hardware due to parallelization or memory bandwidth usage differences.

C2:
  - The adaptive layer freezing proposed in this paper can reduce FLOPs but adds complexity. Calculating the full Fisher Information Matrix (FIM) is computationally intensive because it involves second-order derivatives, which can be prohibitive, especially for large neural networks. The paper must explain why calculating FIM does not introduce additional computation costs.
  - Fisher Information (FI) is calculated based on the current mini-batch. However, the calculation results based on the current mini-batch may not generalize across diverse batches in continual learning. For instance, a layer might appear less informative for one mini-batch but could be essential for others due to shifting data distributions, especially under continual learning settings. Hence, the paper needs to justify this.

C3:
  - The strategy might prioritize rare samples in the dataset but may need to be more informative. This overemphasis can lead to inefficient learning, as these rare samples may not enhance generalization and introduce noise. The paper needs to justify that such a retrieval strategy would maintain the acquired knowledge during training.

**Writing Issues**

  1. Line 071: 'ImageNet)' -> 'ImageNet'.
  2. Line 087: 'ER' has no citation and explanation for the abbreviation.
  3. Fig. 1: No explanation of the y-axis.

---

> ### Author Response · Authors · 2024-11-20
> **Answers to the questions of Reviewer 1YwR (1/2)**
>
> > FLOPs served as a direct measure of computational demand across different models, making it possible to compare CL algorithms fairly. However, it doesn’t account for real-world runtime or memory access patterns, which vary based on hardware and software implementations. For instance, two architectures with similar FLOPs might have different inference times on the same hardware due to parallelization or memory bandwidth usage differences.
>
> $\to$ We agree that FLOPs do not perfectly account for real-world runtime and memory bandwidth. So, as suggested, we report the average wall time over three runs for each experiment and summarize the results in Tab. A below. As shown in the table, performance trends are consistent with those observed in Table 16, where FLOPs were used for comparison, although there is a slight difference due to fluctuations in wall time, which are affected by the factors mentioned above.
>
> While the wall time is a practical measure of real-world performance, it can be affected not only by hardware/software implementation but also by interference from system tasks, making it difficult to measure reproducibly. Out of necessity, we choose to use FLOPs instead.
>
> **Table A. Comparison of Wall time on CIFAR-100 Gaussian setup, with 2 epochs of training per sample encounter**
> | Model | Wall time | Relative wall time compared to ER |
> |----------------------|:---------:|:---------:|
> | ER | 2:17:07 | 1 |
> | DER | 2:29:36 | 1.09 |
> | OCS | 3:59:24 | 1.75 |
> | XDER | 5:16:47 | 2.45 |
> | MEMO | 2:48:24 | 1.23 |
> | CAMA | 2:37:28 | 1.15 |
> | CCL-DC | 9:16:18 | 4.06 |
> | aL-SAR (Ours) | 1:58:45 | 0.86 |
>
> > The adaptive layer freezing proposed in this paper can reduce FLOPs but adds complexity. Calculating the full Fisher Information Matrix (FIM) is computationally intensive because it involves second-order derivatives, which can be prohibitive, especially for large neural networks. The paper must explain why calculating FIM does not introduce additional computation costs.
>
> $\to$ Great point. To avoid the high computational cost in calculating FIM, we use a **first-order approximation** by considering only the diagonal components instead of computing the full FIM, following [1] (in Line 214 and Eq. 2). Since diagonal components are first-order derivatives, which are naturally obtained in gradient descent, aL-SAR’s additional computation costs are negligible compared to the savings achieved by freezing (only 0.00016% increase in computational cost, as shown in Table 16).

---

> ### Author Response · Authors · 2024-11-20
> **Answers to the questions of Reviewer 1YwR (2/2)**
>
> > Fisher Information (FI) is calculated based on the current mini-batch. However, the calculation results based on the current mini-batch may not generalize across diverse batches in continual learning. For instance, a layer might appear less informative for one mini-batch but could be essential for others due to shifting data distributions, especially under continual learning settings. Hence, the paper needs to justify this.
>
> $\to$ Although we compute the FI on the current mini-batch, we **use the exponential moving average (EMA) of the expectations computed from mini-batches over past iterations** to account for the entire dataset (Sec. A.1).  Since EMA estimates continuously incorporate FI of new data, it allows us to account for shifting data distributions while still considering previously encountered data (note that current mini-batch includes both newly encountered data and data from episodic memory).
>
> > Writing Issues - Line 071: 'ImageNet)' -> 'ImageNet'. Line 087: 'ER' has no citation and explanation for the abbreviation. Fig. 1: No explanation of the y-axis.
>
> $\to$ Thank you for the findings! We have corrected typos, added a citation for ER, provided an explanation of the y-axis in Fig. 1, and additionally ran multiple rounds of revision
>
>
> > The strategy might prioritize rare samples in the dataset but may need to be more informative. This overemphasis can lead to inefficient learning, as these rare samples may not enhance generalization and introduce noise. The paper needs to justify that such a retrieval strategy would maintain the acquired knowledge during training.
>
> $\to$ We guess the 'rare sample' you mentioned as the stored samples in episodic memory and of the classes with a few examples. We argue that such rare samples are particularly valuable during training in CL for the following reasons.
>
> Training a model with an imbalanced dataset without accounting for the imbalance would lead to biased predictions and suffer from slow adaptation to novel classes. The biased prediction refers to the samples from minor classes  (e.g., novel classes) are predicted as belonging to major classes (e.g., previously encountered classes) [8, 9]. The slow adaptation leads to poor anytime inference performance [11]. To this end, we prioritized less-trained and forgotten samples to train a model with an imbalanced dataset, inspired by over-sampling (i.e., prioritizing minor classes) [6] and under-sampling (i.e., deprioritizing major classes) [7] strategies from imbalanced setups.
>
> Moreover, our proposed retrieval strategy (i.e., SAR) does not always prioritize rare samples. Specifically, if samples from certain classes are repeatedly used for training while others are not, the use frequency of the unused classes decays continuously through our proposed discounted use frequency. As a result, the priority of unused samples increases, preventing any particular class from being excessively prioritized during training.
>
> [1] Wang et al., SparCL: Sparse Continual Learning on the Edge, NeurIPS 2022
>
> [2] Ghunaim et al., Real-Time Evaluation in Online Continual Learning: A New Hope, CVPR 2023
>
> [3] Csordas et al., SwitchHead: Accelerating Transformers with Mixture-of-Experts Attention, NeurIPS 2024
>
> [4] Chen et al., VanillaNet: the Power of Minimalism in Deep Learning, NeurIPS 2023
>
> [5] Tian et al., Towards Higher Ranks via Adversarial Weight Pruning, NeurIPS 2023
>
> [6] Zhang et al., Learning Fast Sample Re-weighting Without Reward Data, ICCV 2021
>
> [7] Yu et al., Reviving Undersampling for Long-Tailed Learning, arXiv 2024
>
> [8] Caccia et al., New insights on reducing abrupt representation change in online continual learning, ICLR 2021
>
> [9] Seo et al., Learning Equi-angular Representations for Online Continual Learning, CVPR 2024
>
> [10] Koh et al., Online continual learning on class incremental blurry task configuration with anytime inference, ICLR 2022

---

> ### Author Response · Authors · 2024-11-25
> **Discussion Reminder**
>
> We sincerely appreciate your effort in reviewing our submission. We gently remind the reviewer that we tried our best to address your concerns via our replies and manuscript revision. As the discussion period is nearing its end, we would be glad to hear more from you if there are further concerns.

---

> ### Comment · Reviewer_1YwR · 2024-11-26
>
> Thank you for the further clarification. I will raise my score to 8.

---

> > ### Author Response · Authors · 2024-11-26
> > **Thank you for your response**
> >
> > Dear Reviewer 1YwR,
> >
> > Thank you for your comments and for taking the time to review our manuscript.
> >
> > We are happy that our additional discussion and experiments addressed your concerns!
> >
> > If you have any further comments or suggestions, please let us know. We are committed to improving the quality of our work, and we value your feedback.
> >
> > Thank you very much,
> >
> > Authors

---

### Author Response · Authors · 2024-11-20
**General response**

We sincerely thank the reviewers for their valuable feedback and encouraging comments including comprehensive experiments (**1YwR, VVoh, 3EaD**), clear presentation (**j3Dm**), successful application to large models (**VVoh**), suitability and innovative idea (**j3Dm, VVoh, 3EaD**), novel and essential setup (**j3Dm, VVoh**), significant improvements (**VVoh, 3EaD**).

We have uploaded the first revision of the manuscript (changes are highlighted by red color).

---

### Meta-Review · Area_Chair_2nGo · 2024-12-21

**Metareview:**

The paper addresses the challenge of reducing computational overhead and storage requirements in online continual learning (CL). Instead of using training epochs as a measure, it introduces floating-point operations (FLOPs) and total memory size (in bytes) as metrics for computational and memory budgets, respectively. The paper proposes two strategies: similarity-aware retrieval and adaptive layer freezing, which are evaluated through experiments on multiple benchmark datasets.

**Strengths**

- Using FLOPs and total memory size as metrics offers a fair and standardized way to compare different online CL algorithms.
- The proposed adaptive layer freezing and similarity-aware retrieval strategies are effective for computationally efficient online CL, as demonstrated by experimental results.

**Weaknesses**

- Reviewers raised concerns about the additional computational overhead introduced by Fisher information and gradient similarity calculations.
- The impact of the proposed strategies on catastrophic forgetting, a critical aspect of CL, is not clearly established in the paper.

**Overall Assessment**
- The authors’ rebuttal included additional experiments and convincing responses, which adequately addressed the major concerns raised during the review process. The paper provides a meaningful contribution to computationally efficient online CL and has the potential to inspire further research in the field.

**Additional Comments On Reviewer Discussion:**

All reviewers actively participated in discussions with the authors during the rebuttal phase. The authors provided new results, which significantly improved the quality of the revised paper. As a result, multiple reviewers raised their scores, leading to a consensus to accept the paper.

---

### Decision · Program_Chairs · 2025-01-22

Accept (Spotlight)